# PiSSA: Principal Singular Values and Singular Vectors Adaptation of Large Language Models

**Fanxu Meng**[1,2]**, Zhaohui Wang**[1]**, Muhan Zhang**[1,2*]
[1]Institute for Artificial Intelligence, Peking University
[2]State Key Laboratory of General Artificial Intelligence, Peking University
https://github.com/GraphPKU/PiSSA

## Abstract

To parameter-efficiently fine-tune (PEFT) large language models (LLMs), the low-rank adaptation (LoRA) method approximates the model changes $\Delta W \in \mathbb{R}^{m \times n}$ through the product of two matrices $A \in \mathbb{R}^{m \times r}$ and $B \in \mathbb{R}^{r \times n}$, where $r \ll \min(m, n)$, $A$ is initialized with Gaussian noise, and $B$ with zeros. LoRA **freezes the original model** $W$ and **updates the "Noise & Zero" adapter**, which may lead to slow convergence. To overcome this limitation, we introduce **Pri**ncipal **S**ingular values and **S**ingular vectors **A**daptation (PiSSA). PiSSA shares the same architecture as LoRA, but initializes the adaptor matrices $A$ and $B$ with the principal components of the original matrix $W$, and put the remaining components into a residual matrix $W^{res} \in \mathbb{R}^{m \times n}$ which is frozen during fine-tuning. Compared to LoRA, PiSSA **updates the principal components** while **freezing the "residual" parts**, allowing faster convergence and enhanced performance. Comparative experiments of PiSSA and LoRA across 11 different models, ranging from 184M to 70B, encompassing 5 NLG and 8 NLU tasks, reveal that PiSSA consistently outperforms LoRA under identical experimental setups. On the GSM8K benchmark, Gemma-7B fine-tuned with PiSSA achieves an accuracy of 77.7%, surpassing LoRA's 74.53% by 3.25%. Due to the same architecture, PiSSA is also compatible with quantization to further reduce the memory requirement of fine-tuning. Compared to QLoRA, QPiSSA (PiSSA with 4-bit quantization) exhibits smaller quantization errors in the initial stages. Fine-tuning LLaMA-3-70B on GSM8K, QPiSSA attains an accuracy of 86.05%, exceeding the performance of QLoRA at 81.73%. Leveraging a fast SVD technique, PiSSA can be initialized in only a few seconds, presenting a negligible cost for transitioning from LoRA to PiSSA.

## 1 Introduction

Fine-tuning large language models (LLMs) is a highly effective technique for boosting their capabilities in various tasks [1, 2, 3, 4], ensuring models to follow instructions [5, 6, 7], and instilling models with desirable behaviors while eliminating undesirable ones [8, 9]. However, the fine-tuning process for very large models is accompanied by prohibitive costs. For example, regular 16-bit fine-tuning of a LLaMA 65B parameter model requires over 780 GB of GPU memory [10], and the VRAM consumption for training GPT-3 175B reaches 1.2TB [11]. Consequently, various parameter-efficient fine-tuning (PEFT) [12, 13] methods have been proposed to reduce the number of parameters and memory usage required for fine-tuning. Due to the ability to maintain the performance of full fine-tuning without adding additional inference latency, Low-Rank Adaptation (LoRA) [11] has emerged as a popular PEFT method.

---

*Correspondence to: Muhan Zhang <muhan@pku.edu.cn>

38th Conference on Neural Information Processing Systems (NeurIPS 2024).

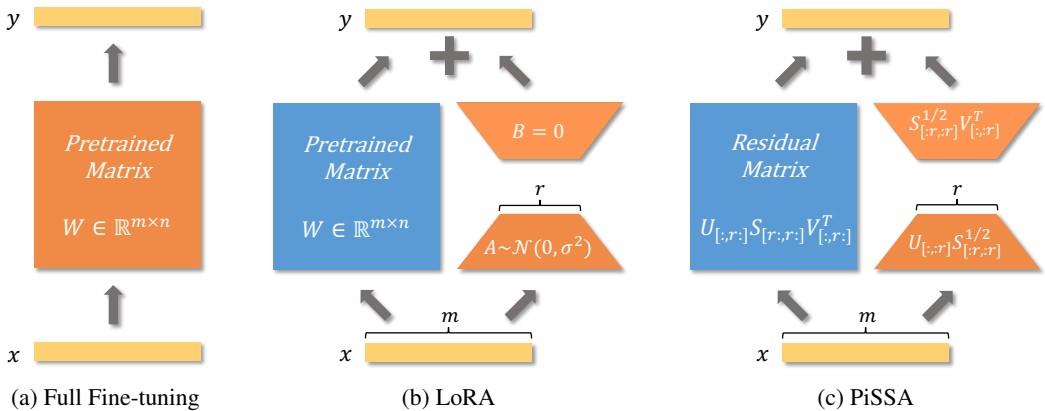

(a) Full Fine-tuning        (b) LoRA        (c) PiSSA

Figure 1: The comparison among Full Fine-tuning, training with LoRA, and PiSSA. In this visualization, blue modules represent parts of the model whose parameters are frozen during training, while orange modules indicate components that require updates. QLoRA quantizes the pretrained matrix in LoRA to 4-bit, whereas QPiSSA quantizes the residual matrix in PiSSA.

Table 1: Comparison of similarities and differences between PiSSA and LoRA. In this table, **bold** highlights the model's primary component, while underline denotes the residual component.

| | LoRA | PiSSA |
|---|---|---|
| Forward | $Y = X(\mathbf{W} + \underline{\Delta W}) = X(\mathbf{W} + \underline{AB})$ | $Y = X(\underline{W^{res}} + \mathbf{W^{pri}}) = X(\underline{W^{res}} + \mathbf{AB})$ |
| Initialization | $\underline{A} \sim \mathcal{N}(0, \sigma^2) \in \mathbb{R}^{m \times r}$ 

 $\underline{B} = 0 \in \mathbb{R}^{r \times n}$ | $\mathbf{A} = U_{[:,:\mathbf{r}]} S_{[:\mathbf{r},:\mathbf{r}]}^{1/2} \in \mathbb{R}^{m \times r}$ 
 $\mathbf{B} = S_{[:\mathbf{r},:\mathbf{r}]}^{1/2} V_{[:,:\mathbf{r}]}^{\top} \in \mathbb{R}^{r \times n}$ 
 $\underline{W^{res}} = U_{[:,\underline{r}:]} S_{[\underline{r}:,\underline{r}:]} V_{[:,\underline{r}:]}^{\top} \in \mathbb{R}^{m \times n}$ |
| Gradient | $\frac{\partial L}{\partial A} = X^{\top} \left( \frac{\partial L}{\partial Y} \right) \underline{B}^{\top} \to \underline{0}$ 
 $\frac{\partial L}{\partial B} = \underline{A}^{\top} X^{\top} \left( \frac{\partial L}{\partial Y} \right) \to \underline{\text{Random Direction}}$ | $\frac{\partial L}{\partial A} = X^{\top} \left( \frac{\partial L}{\partial Y} \right) \mathbf{B}^{\top} \to \mathbf{Principal}$ 
 $\frac{\partial L}{\partial B} = \mathbf{A}^{\top} X^{\top} \left( \frac{\partial L}{\partial Y} \right) \to \mathbf{Principal}$ |
| Comparison | Fine-tunes noise while freezing $\mathbf{W}$. 
 Slow convergence and underperformance. 
 QLoRA cannot reduce quantization error. | Fine-tunes **principal** parts freezing $\underline{W^{res}}$. 
 **Fast** convergence and **better** performance. 
 QPiSSA **can** reduce quantization error. |

LoRA [11] hypothesizes that the modifications to parameter matrices during fine-tuning exhibit low-rank properties. As depicted in Figure 1b, for a pre-trained weight matrix $W \in \mathbb{R}^{m \times n}$, LoRA substitutes the updates with a low-rank decomposition $\Delta W = AB$, where $A \in \mathbb{R}^{m \times r}$ and $B \in \mathbb{R}^{r \times n}$, and the rank $r \ll \min(m, n)$. For $Y = XW$, the modified forward pass is as follows:

$$Y = X(W + \Delta W) = X(W + AB), \tag{1}$$

A random Gaussian initialization is used for $A$ and zero for $B$, making $AB = 0$ at the beginning of training, thereby the injection of adapters does not affect the model's output initially. LoRA avoids the need to compute gradients or maintain the optimizer states for the original matrix $W$, instead optimizing the injected, significantly smaller low-rank matrices $A, B$. Thus, it could reduce the number of trainable parameters by $10,000\times$ and the GPU memory requirement by $3\times$ [11]. LoRA is capable of achieving comparable performance to full parameter fine-tuning. By integrating the quantization of pre-trained matrices $W$, LoRA also enables reducing the average memory requirements by $16\times$ [10]. Meanwhile, the adapters can still utilize higher precision weights, thus, the quantization usually does not significantly degrade the performance of LoRA.

According to Equation 1, the gradients of A and B are $\frac{\partial L}{\partial A} = X^{\top} \left( \frac{\partial L}{\partial Y} \right) B^{\top}$ and $\frac{\partial L}{\partial B} = A^{\top} X^{\top} \left( \frac{\partial L}{\partial Y} \right)$. Compared to full fine-tuning, using LoRA initially does not change the output $Y$ for the same input $X$, so the magnitude and direction of gradient are primarily determined by the values of $A$ and $B$. Since $A$ and $B$ are initialized with Gaussian noise and zeros in LoRA, the gradients could be small

and uninformative for a long time, leading to slow convergence in the fine-tuning process. We also observe this phenomenon empirically, as LoRA often wastes much time around the initial point.

Our **Pri**ncipal **S**ingular values and **S**ingular vectors **A**dapter (PiSSA) diverges from LoRA and its successors by focusing not on approximating $\Delta W$, but $W$. We apply singular value decomposition (SVD) to matrix $W$. Based on the magnitude of the singular values, we partition $W$ into two parts: the principal low-rank matrix $W^{pri}$, comprising a few largest singular values, and the residual matrix $W^{res}$, which possesses the remaining smaller singular values (with a larger quantity, representing a possible long-tail distribution). The principal matrix $W^{pri}$ can be represented by the product of $A \in \mathbb{R}^{m \times r}$ and $B \in \mathbb{R}^{r \times n}$, where $r \ll \min(m, n)$. As depicted in Figure 1c, $A$ and $B$ are initialized based on the principal singular values and singular vectors and are trainable. Conversely, $W^{res}$ is initialized with the product of the residual singular values and singular vectors and remains frozen during fine-tuning. Since the principal singular vectors represent the directions in which the matrix $W$ has the most significant stretching or impact, by directly tuning these principal components, PiSSA is able to **fit the training data faster and better** (as demonstrated in Figure 2a). Moreover, the loss and gradient norm curves of PiSSA often demonstrate a similar trend to those of full parameter fine-tuning in our experiments (Figure 4), indicating that fine-tuning the principal components matches the behavior of fine-tuning the full matrix to some degree.

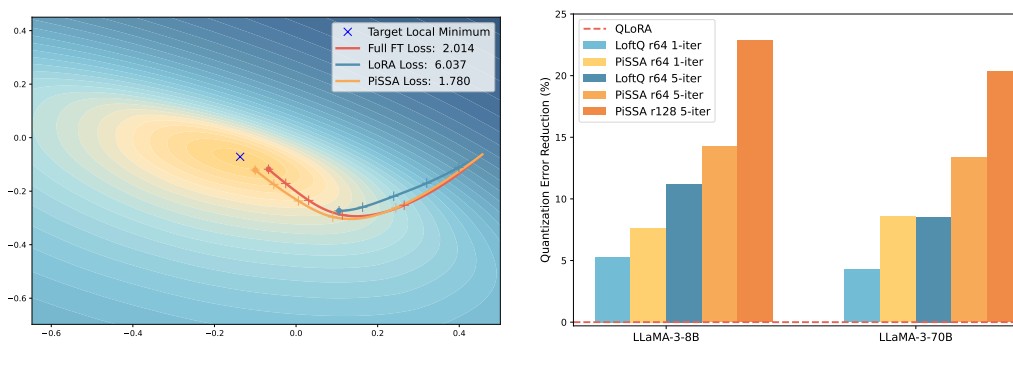

(a) PiSSA converges more rapidly.       (b) PiSSA reduces more quantization error.

Figure 2: We illustrate the two key advantages of PiSSA: converging faster and better, and reducing quantization error. In the left figure, we use a toy example to show PiSSA's faster convergence, where we first train a two-layer MLP classifying odd numbers of MNIST, and then fine-tune the model on even numbers. PiSSA finds the right direction more quickly and achieves a lower loss with the same number of steps. In the right figure, PiSSA reduces quantization error more effectively than LoftQ [14], with an optional 5-iteration SVD for further error reduction, as detailed in Appendix E.

Because the principal components $W^{pri}$ are preserved in the adapter at full precision, an additional benefit of PiSSA is that when applying quantization to the frozen part $W^{res}$, we can significantly **reduce the quantization error** compared to QLoRA (which quantizes the entire $W$), as illustrated in Figure 2b. Therefore, PiSSA is even more compatible with quantization than LoRA, making it a superior plug-and-play substitution for LoRA.

Our paper makes several significant contributions:

- We analyze the initial gradient magnitude and direction in LoRA, demonstrating that $A$ initially has a zero gradient and $B$ has a random gradient, which slows down convergence and may lead to convergence at suboptimal local minima.

- We propose PiSSA initialization, a novel method that approximates the optimization direction of full-parameter fine-tuning by adapting a model's principal components. To our knowledge, PiSSA is the first to apply SVD to the original model, using principal singular values and vectors to initialize the adapter for fine-tuning, while keeping the residual components frozen. Experiments show that PiSSA converges faster and outperforms LoRA.

- We combine PiSSA with NF4 quantization to propose QPiSSA, which reduces quantization error by about 20% compared to QLoRA, while maintaining the fast convergence and high performance of PiSSA.

## 2 Related Works

The vast complexity and computational needs of large language models (LLMs) with billions of parameters present significant hurdles in adapting them for specific downstream tasks. Parameter Efficient Fine-Tuning (PEFT) [12, 13] emerges as a compelling solution by minimizing the fine-tuning parameters and memory requirements while achieving comparable performance to full fine-tuning. PEFT encompasses strategies like partial fine-tuning [15, 16, 17, 18, 19, 20, 21, 22], soft prompt fine-tuning [23, 24, 25, 26, 27, 28, 29], non-linear adapter fine-tuning [30, 31, 32, 33, 34, 35, 36, 37, 38, 39], and low rank adapter based fine-tuning [40, 41, 11, 42].

LoRA [11] injects trainable adapters to the linear layers. After fine-tuning, these adaptations can be re-parameterized into the standard model structure, thus gaining widespread adoption due to their ability to maintain the model's original architecture while enabling efficient fine-tuning. Following LoRA, AdaLoRA [42] dynamically learns the rank size needed for LoRA in each layer of the model. DeltaLoRA [43] updates the original weights of the model using parameters from adapter layers, enhancing LoRA's representational capacity. LoSparse [44] incorporates LoRA to prevent pruning from eliminating too many expressive neurons. DoRA [45] introduces a magnitude component to learn the scale of $\Delta W$ while utilizing the original AB as a direction component of $\Delta W$. Unlike LoRA and its successors, which focus on learning low-rank approximations of weight updates, our PiSSA directly tunes the essential low-rank parts of the model while keeping the noisier, high-rank, and nonessential parts frozen. Although our approach differs in philosophy from LoRA, it shares most of LoRA's structural benefits and can be extended by these methods to enhance its performance.

QLoRA [10] integrates LoRA with 4-bit NormalFloat (NF4) quantization, along with Double Quantization and Paged Optimizers, enabling the fine-tuning of a 65B parameter model on a single 48GB GPU while preserving the performance of full 16-bit fine-tuning tasks. QA-LoRA [46] introduces group-wise operators to increase the degree of freedom in low-bit quantization. LoftQ [14] reduces quantization error by decomposing the quantization error matrix of QLoRA and retaining the principal components with an adapter. PiSSA can also be combined with quantization techniques, and we have found that PiSSA significantly reduces quantization error compared to QLoRA and LoftQ.

## 3 PiSSA: Principal Singular Values and Singular Vectors Adaptation

This section formally presents our **Pri**ncipal **S**ingular values and **S**ingular vectors **A**daptation method. PiSSA computes the singular value decomposition (SVD) of matrices $W$ within the self-attention and multilayer perceptron (MLP) layers. The (economy size) SVD of a matrix $W \in \mathbb{R}^{m \times n}$ is given by $W = USV^\top$, where $U \in \mathbb{R}^{m \times \min(m,n)}, V \in \mathbb{R}^{n \times \min(m,n)}$ are the singular vectors with orthonormal columns, and $V^\top$ is the transpose of $V$. $S = \mathrm{diag}(\mathbf{s}) \in \mathbb{R}^{\min(m,n) \times \min(m,n)}$, where the operation $\mathrm{diag}(\mathbf{s})$ transforms $\mathbf{s}$ to a diagonal matrix $S$, and $\mathbf{s} \in \mathbb{R}_{\geq 0}^{\min(m,n)}$ represents the singular values arranged in descending order. When the top $r$ singular values $\mathbf{s}_{[:r]}$ are significantly larger than the remaining singular values $\mathbf{s}_{[r:]}$, we denote the intrinsic rank of $W$ as $r$. Consequently, $S$, along with $U$ and $V$, can be divided into two groups: the principal singular values and vectors—$\{U_{[:,:r]}, S_{[:r,:r]}, V_{[:,:r]}\}$, and the residual singular values and vectors—$\{U_{[:,r:]}, S_{[r:,r:]}, V_{[:,r:]}\}$, where the matrix slicing notations are the same as those in PyTorch and $[: r]$ denotes the first $r$ dimensions. The principal singular values and vectors are utilized to initialize the injected adapter consisting of $A \in \mathbb{R}^{m \times r}$ and $B \in \mathbb{R}^{r \times n}$:

$$A = U_{[:,:r]} \, S_{[:r,:r]}^{1/2} \in \mathbb{R}^{m \times r}, \tag{2}$$

$$B = S_{[:r,:r]}^{1/2} \, V_{[:,:r]}^\top \in \mathbb{R}^{r \times n}. \tag{3}$$

The residual singular values and vectors are used to build the residual matrix which is frozen during fine-tuning:

$$W^{res} = U_{[:,r:]} \, S_{[r:,r:]} \, V_{[:,r:]}^\top \in \mathbb{R}^{m \times n}. \tag{4}$$

As indicated by Equation 5, the integration of $AB$ with the residual matrix also preserves the full capability of the pre-trained model in the beginning of fine-tuning:

$$Y = XW = X(W^{res} + W^{pri}) = X(W^{res} + AB). \tag{5}$$

Similar to LoRA, the gradients of $A$ and $B$ are also given by $\frac{\partial L}{\partial A} = X^\top \left( \frac{\partial L}{\partial Y} \right) B^\top$ and $\frac{\partial L}{\partial B} = A^\top X^\top \left( \frac{\partial L}{\partial Y} \right)$. Since elements of $\mathbf{s}_{[:r]} \gg$ elements of $\mathbf{s}_{[r:]}$, the trainable adapter $W^{pri} = AB$ contains the most essential directions of $W$. In the ideal case, training $AB$ mirrors the process of fine-tuning the entire model despite using fewer parameters. The ability to directly fine-tune the most essential part of a model enables PiSSA to converge faster and better. In contrast, LoRA initializes the adapters $A$ and $B$ with Gaussian noise and zeros while keeping $W$ frozen. Consequently, the gradients are small or in random directions during the early stages of fine-tuning, possibly introducing much waste of gradient descent steps. Moreover, an inferior initialization might lead to suboptimal local minimum, resulting in worse generalization performance.

Since PiSSA shares the identical architecture with LoRA, it inherits most of LoRA's benefits. These include but are not limited to the capability of fine-tuning a model with a reduced number of trainable parameters, quantizing the residual model to decrease memory consumption during forward propagation in training, and easy deployment. The adapter's straightforward linear structure facilitates the integration of trainable matrices with the pre-trained weights upon deployment, thereby maintaining the original inference speed of a fully fine-tuned model. Employing the Fast SVD technique [47] allowed PiSSA to finish initialization in several seconds (Appendix B), which is a negligible cost.

For storage efficiency, we can choose not to store the dense parameter matrix $\Delta W$, but to store the low-rank matrices, $\Delta A$ and $\Delta B$ instead. As shown in Appendix C, leveraging solely the $\Delta A$ and $\Delta B$ facilitates their seamless integration with the original pre-trained models. Finally, one pre-trained model can accommodate multiple $\Delta A, \Delta B$, fine-tuned by diverse PiSSA or LoRA procedures, which enables fast adaptation of the pre-trained model to different downstream applications.

## 4 QPiSSA: An Extension Method with Lower Quantization Error

Quantization divides the value range of a matrix into several continuous regions, and maps all values falling inside a region into the same "quantized" value. It is an effective technique to reduce the memory consumption of forward propagation. At the same time, LoRA greatly reduces the backward memory requirement, making it highly suitable to use LoRA and quantization together, where the base model is quantized for memory-efficient forward propagation, and the LoRA adaptors are kept in full precision for accurate backward parameter updates. One representative previous work, QLoRA, quantizes the base model to Normal Float 4-bit (NF4) and initializes the full-precision $A$ and $B$ with Gaussian-Zero initialization. Therefore, the overall error is given by:

$$\text{Quantization Error of QLoRA} = ||W - (nf4(W) + AB)||_* = ||W - nf4(W)||_*, \quad (6)$$

where $||M||_*$ denotes the nuclear norm (also known as the trace norm [48]), defined as:

$$||M||_* = \text{trace} \left( \sqrt{M^*M} \right) = \sum_{i=1}^{\min\{m,n\}} \sigma_i(M), \quad (7)$$

where $\sigma_i(M)$ is the $i^{\text{th}}$ singular value of $M$. As we can see, the quantization error of QLoRA is the same as that of directly quantizing the base model. Our QPiSSA, however, **does not quantize the base model but the residual model**. Therefore, its error is given by:

$$\text{Quantization Error of QPiSSA} = ||W - (nf4(W^{res}) + AB)||_* = ||W^{res} - nf4(W^{res})||_*. \quad (8)$$

Since the residual model has removed the large-singular-value components, $W^{res}$ has a **narrower distribution** than that of $W$, as can be seen in Figures 3a and 3b (comparing the singular value distributions of $W$ and $W^{res}$), as well as Figures 3c and 3f (comparing the value distributions of $W$ and $W^{res}$), which is **beneficial for reducing the quantization error**. Moreover, given that NF4 is optimized for data with a normal distribution, as discussed by Dettmers et al. [10], we fit the values of $W$ and $W^{res}$ to a Gaussian distribution respectively. As illustrated in Figures 3c and 3f, $W^{res}$ aligns more closely with a Gaussian distribution and exhibits a smaller standard deviation, making it more suitable for applying NF4 than $W$. Both the above lead QPiSSA to achieve a significantly lower quantization error than QLoRA, shown in Figures 3d and 3e.

Besides the advantage of reducing quantization error, QPiSSA's gradient direction is similar to that of PiSSA, resulting in significantly better fine-tuning performance compared to QLoRA.

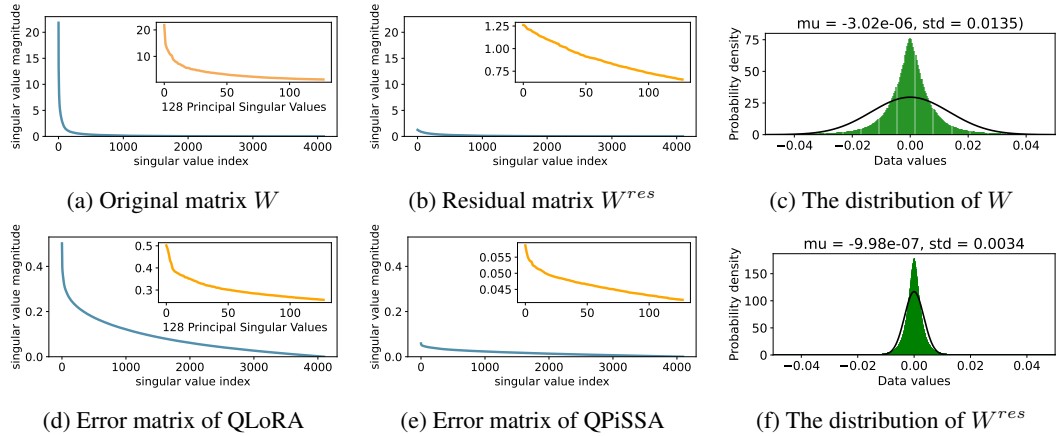

(a) Original matrix $W$       (b) Residual matrix $W^{res}$       (c) The distribution of $W$

(d) Error matrix of QLoRA      (e) Error matrix of QPiSSA      (f) The distribution of $W^{res}$

Figure 3: Visualizations of LLaMA 2-7B's "layers[0].self_attn.q_proj" matrix, with distributions for the full model shown in Appendix G. Figures (a), (b), (d), and (e) display the singular values of $W, W^{res}, W - nf4(W)$, and $W^{res} - nf4(W^{res})$, respectively. Figures (c) and (f) show the data distributions of $W$ and $W^{res}$.

## 5 Experiments

The experiments were conducted on the NVIDIA A800-SXM4(80G) GPU. In our experiments, we adopt the Alpaca [49] implementation strategy, using the AdamW optimizer with a batch size of 128, a learning rate of 2e-5, cosine annealing schedules, and a warmup ratio of 0.03, without any weight decay. As discussed in Section B.3 of QLoRA [10], we compute the loss using only the responses from the instruction-following datasets. We ensure lora_alpha is always equal to lora_r, set lora_dropout to 0, and incorporate the adapters into all linear layers of the base model. We utilize the Float32 computation type for both the base model and the adapter in LoRA and PiSSA. For QLoRA, LoftQ, and QPiSSA, we use 4-bit NormalFloat [10] for the base model and Float32 for the adapter. BFloat16 [50] is used for full parameter fine-tuning to save the resources (see Appendix D).

### 5.1 Evaluating the Performance of PiSSA on both NLG and NLU Tasks

We begin by comparing PiSSA, LoRA, and full-parameter fine-tuning on natural language generation (NLG) tasks. We fine-tuned LLaMA 2-7B [51], Mistral-7B-v0.1 [52], and Gemma-7B [53] on the MetaMathQA dataset [2] to assess their mathematical problem-solving capabilities on the GSM8K [54] and MATH [2] validation sets. Additionally, the models were fine-tuned on the Code-Feedback dataset [55] and evaluated for coding proficiency using the HumanEval [56] and MBPP [57] datasets. Furthermore, the models were trained on the WizardLM-Evol-Instruct dataset [7] and tested for conversational abilities on the MT-Bench dataset [6]. All experiments were conducted using subsets containing 100K data points and were trained for only one epoch to reduce training overhead.

As shown in Table 2, across all models and tasks, fine-tuning with PiSSA consistently surpasses the performance of fine-tuning with LoRA. Further experiments demonstrated that this improvement is robust across various amounts of training data and epochs (Section 5.2), including both 4-bit and full precision (Section 5.3), different model sizes and types (Section 5.4), and varying proportions of trainable parameters (Section 5.5).

We also evaluate PiSSA's natural language understanding (NLU) capability on the GLUE benchmark [59] with DeBERTa-v3-base [60]. Table 3 presents the results across 8 tasks. PiSSA outperforms LoRA in 7 out of 8 NLU tasks, achieving an overall average improvement of 1.21%. Upon reviewing the training loss on the exceptional dataset, MNLI, we observed that PiSSA's average loss of $0.17$ was lower than LoRA's $0.24$ in the final epoch. This indicates that the fitting ability of PiSSA remains stronger than that of LoRA.

Table 2: Comparison of PiSSA and LoRA on NLG tasks, with results averaged over three runs and reported with standard deviations.

| Model | Strategy | GSM8K | MATH | HumanEval | MBPP | MT-Bench |
|-------|----------|-------|------|-----------|------|----------|
| LLaMA 2-7B | Full FT | 49.13±0.21 | 7.29±0.22 | 21.20±0.30 | 35.59±0.25 | **4.91±0.01** |
| | LoRA(gaussian) | 42.85±0.12 | 5.50±0.33 | 18.35±0.31 | 35.50±0.14 | 4.59±0.07 |
| | LoRA(kaiming) | 43.23±0.64 | 5.90±0.16 | 18.21±0.45 | 35.47±0.37 | 4.56±0.04 |
| | PiSSA | **53.22±0.55** | **7.47±0.34** | **21.92±0.38** | **37.24±0.63** | 4.88±0.03 |
| Mistral-7B | Full FT | 69.91±0.25 | 18.64±0.35 | 45.31±0.14 | 51.46±0.13 | 4.95±0.05 |
| | LoRA(gaussian) | 69.50±0.42 | 20.08±0.20 | 43.78±0.34 | 58.46±0.37 | 4.90±0.05 |
| | LoRA(kaiming) | 69.40±0.25 | 19.99±0.44 | 43.74±0.14 | 58.39±0.02 | 4.93±0.05 |
| | PiSSA | **73.31±0.23** | **23.12±0.52** | **46.88±0.25** | **62.55±0.58** | **5.34±0.04** |
| Gemma-7B | Full FT | 72.09±0.32 | 22.71±0.34 | 47.02±0.27 | 55.67±0.50 | 5.40±0.12 |
| | LoRA(gaussian) | 75.11±0.64 | 30.41±0.48 | 53.70±0.23 | 65.58±0.29 | 4.98±0.02 |
| | LoRA(kaiming) | 74.53±0.47 | 29.90±0.16 | 53.57±0.27 | 65.25±0.29 | 4.97±0.09 |
| | PiSSA | **77.78±0.32** | **31.33±0.33** | **54.31±0.28** | **66.17±0.43** | **5.64±0.10** |

Table 3: Comparison of PiSSA and LoRA on NLU tasks. $\text{LoRA}^G$ and $\text{LoRA}^K$ denote LoRA with Gaussian and Kaiming initialization for $B$, respectively. Results for full fine-tuning, BitFit [15], HAdapter [30], PAdapter [36], $\text{LoRA}^G$ [11] and AdaLoRA are from AdaLoRA [58], averaged over five runs. Remaining methods are averaged over three runs, with details in Appendix L.

| Method | Params | MNLI | SST2 | MRPC | CoLA | QNLI | QQP | RTE | STSB | ALL |
|--------|--------|------|------|------|------|------|-----|-----|------|-----|
| Full FT | 184M | 89.90 | 95.63 | 89.46 | 69.19 | 94.03 | **92.40** | 83.75 | 91.60 | 88.25 |
| BitFit | 0.1M | 89.37 | 94.84 | 87.75 | 66.96 | 92.24 | 88.41 | 78.70 | 91.35 | 86.20 |
| HAdapter | 1.22M | 90.13 | 95.53 | 89.95 | 68.64 | 94.11 | 91.91 | 84.48 | 91.48 | 88.28 |
| PAdapter | 1.18M | 90.33 | 95.61 | 89.46 | 68.77 | 94.29 | 92.04 | 85.20 | 91.54 | 88.41 |
| $\text{LoRA}^G$ | 1.33M | 90.65 | 94.95 | 89.95 | 69.82 | 93.87 | 91.99 | 85.20 | 91.60 | 88.50 |
| $\text{LoRA}^K$ | 1.33M | 89.96 | 95.64 | 90.28 | 70.69 | 93.84 | 92.03 | 84.84 | 91.68 | 88.62 |
| DoRA | 1.27M | 90.29 | 95.79 | 90.93 | 70.85 | 94.10 | 92.07 | 86.04 | 91.79 | 88.98 |
| AdaLoRA | 1.27M | **90.76** | 96.10 | 90.69 | 71.45 | **94.55** | 92.23 | 88.09 | 91.84 | 89.46 |
| PiSSA | 1.33M | 90.37 | **96.22** | **91.50** | **73.12** | 94.43 | 92.33 | **88.69** | **92.00** | **89.83** |

## 5.2 Experiments using Full Data and More Epochs

In this section, we finetune LLaMA 2-7B model on the complete MetaMathQA-395K dataset for 3 epochs to ensure thorough saturation. The training loss and gradient norms is visualized to demonstrate quicker convergence and evaluated on the GSM8K dataset every 1000 steps to demonstrate superior performance of PiSSA compared to LoRA. The results are depicted in Figure 4. Additionally, similar comparisons on Mistral-7B and Gemma-7B are detailed in Appendix J.

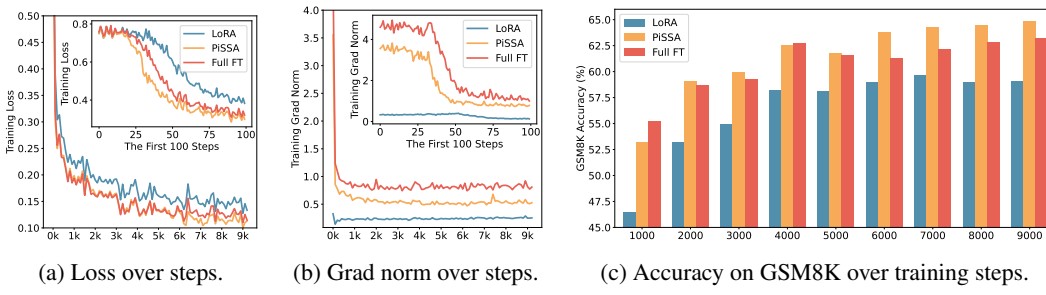

(a) Loss over steps.      (b) Grad norm over steps.      (c) Accuracy on GSM8K over training steps.

Figure 4: The loss, grad norm, and evaluation accuracy over the training steps of LoRA (indicated in blue), PiSSA (in orange), and full parameter fine-tuning (in red).

According to Figure 4a, the loss of PiSSA reduces rapidly during the first 100 steps, and the grad norm (shown in Figure 4b) of PiSSA is significantly higher than that of LoRA, with a trend similar to full fine-tuning. Throughout the process, the loss of PiSSA remains lower than that of LoRA, indicating

that PiSSA converges to a better local optimum. As shown in Figure 4c, PiSSA consistently achieves higher accuracy compared to LoRA, and in most cases also surpasses full parameters fine-tuning. We hypothesize that this is because PiSSA is a denoised version of full fine-tuning. Comparing the grad norm and loss curves of PiSSA and full fine-tuning, we can see that the larger grad norm of full fine-tuning does not bring lower loss, indicating that a portion of the grad norm is spent on noisy directions not beneficial for loss reduction. This phenomenon is consistent with Figure 2a.

## 5.3 Conducting 4-bit Quantization Experiments

In this section, we first compare the initial quantization error reduction ratio of PiSSA, QLoRA, and LoftQ. This ratio is defined as $(1 - \frac{||W-(nf4(W^{'})+AB)||_*}{||W-nf4(W)||_*}) \times 100\%$, measuring the relative error decrease achieved by each mehod compared to directly quantizing the base model. The partial results are presented in Table 4, and the complete results can be found in Table 8 in Appendix E.

Table 4: The quantization error reduction ratio of QLoRA, LoftQ, and PiSSA across different layers.

|  | Method | Rank | Q | K | V | O | Gate | Up | Down | AVG |
|---|---|---|---|---|---|---|---|---|---|---|
| LLaMA 2-7B | QLoRA | – | 0 | 0 | 0 | 0 | 0 | 0 | 0 | 0 |
|  | loftQ | 128 | 16.5 | 16.5 | 15.9 | 16.0 | 12.4 | 12.4 | 12.3 | 14.6 |
|  | **PiSSA** | **128** | **27.9** | **27.2** | **18.7** | **18.6** | **15.8** | **13.6** | **13.6** | **19.4** |
| LLaMA 3-8B | QLoRA | – | 0 | 0 | 0 | 0 | 0 | 0 | 0 | 0 |
|  | loftQ | 128 | 16.4 | 29.8 | 28.8 | 16.1 | 11.9 | 11.7 | 11.7 | 18.1 |
|  | **PiSSA** | **128** | **26.3** | **41.7** | **32.3** | **20.1** | **14.4** | **12.5** | **12.9** | **22.9** |
| LLaMA 3-70B | QLoRA | – | 0 | 0 | 0 | 0 | 0 | 0 | 0 | 0 |
|  | LoftQ | 64 | 6.1 | 17.8 | 17.0 | 6.0 | 4.3 | 4.4 | 4.2 | 8.5 |
|  | **PiSSA** | **64** | **15.7** | **34.2** | **18.9** | **7.5** | **6.7** | **5.7** | **4.7** | **13.4** |
|  | **PiSSA** | **128** | **23.2** | **49.0** | **30.5** | **12.5** | **10.1** | **8.8** | **8.2** | **20.3** |

In Table 4, PiSSA reduces the quantization error by about 20% compared to directly quantizing the base model. The reduction is more significant for lower-rank matrices. For instance, in the LLaMA-3-70B [61], all "Key" projection layers see a reduction of 49%. The results in Table 4 validate that QLoRA, discussed in Section 4, does not reduce quantization error. In contrast, PiSSA significantly outperforms LoftQ in reducing quantization error, as further discussed in Appendix H.

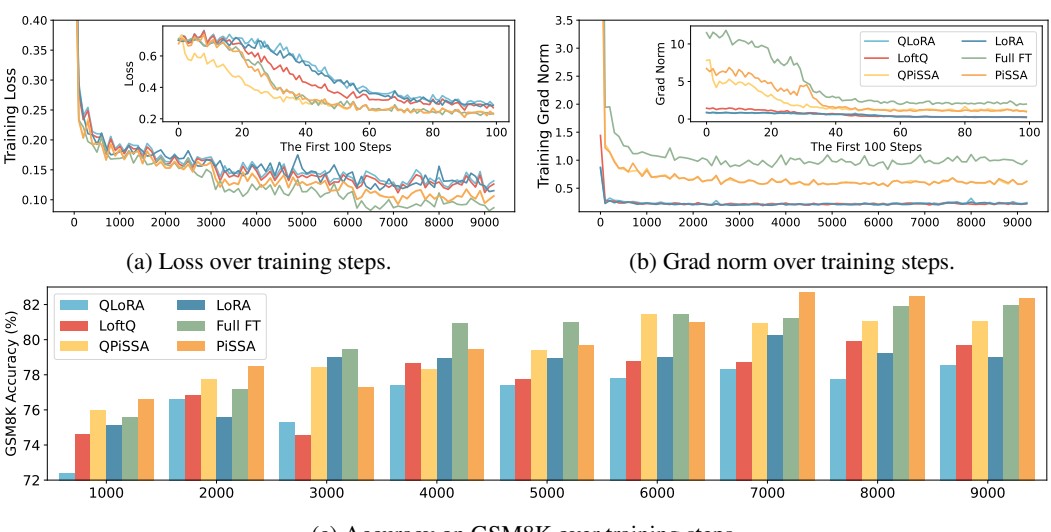

(a) Loss over training steps.    (b) Grad norm over training steps.

(c) Accuracy on GSM8K over training steps.

Figure 5: The loss, grad norm, and evaluation accuracy over the training steps of (Q)LoRA, (Q)PiSSA, LoftQ and full parameter fine-tuning.

The difference between QPiSSA and PiSSA is the quantization of the residual model to 4 bits. As introduced in Section 4, the residual model has less influence on the optimal direction compared with the principal adapter, which is the same in both QPiSSA and PiSSA. Therefore, besides reducing the quantization error, we expect QPiSSA to also converge faster than QLoRA and LoftQ. We train LLaMA 3-8B using LoRA/QLoRA, PiSSA/QPiSSA, LoftQ, and full fine-tuning on MetaMathQA-395K for 3 epochs, recording the loss, gradient norm, and accuracy on GSM8K, as shown in Figure 5.

According to Figure 5, QPiSSA's loss reduction speed in the first 100 steps is even faster than PiSSA and full fine-tuning. Although LoftQ can reduce the quantization error, its loss convergence speed is not faster than LoRA and QLoRA, indicating that QPiSSA's ability to reduce the quantization error and its fast convergence might also be orthogonal capabilities. After sufficient training, QPiSSA's loss is also much lower than that of LoRA/QLoRA and LoftQ. The grad norm is significantly larger than those of LoRA/QLoRA and LoftQ. In terms of fine-tuning performance, QPiSSA's accuracy is higher than that of QLoRA and LoftQ and even better than that of full-precision LoRA.

## 5.4 Experiments Across Various Sizes and Types of Models

In this section, we compare (Q)PiSSA and (Q)LoRA across 9 models, ranging from 7-70B parameters, including LLaMA 2-7/13B [51], LLaMA-3-8/70B [61], Mistral-7B [52], Gemma-7B [53], and Qwen1.5-7B [62], Yi-1.5-34B [63] and MoE models: DeepSeek-MoE-16B [64] and Mixtral-8x7B [65]. These models were fine-tuned on the MetaMathQA-100K and CodeFeedback-100K dataset and evaluated on the GSM8K and HumanEval. DeepSeek-MoE-16B, Mixtral-8x7B, Yi-1.5-34B, and LLaMA-3-70B were fine-tuned with QPiSSA and QLoRA, while the other models were using PiSSA and LoRA. From Figure 6, (Q)PiSSA, compared to (Q)LoRA, shows improved accuracy across various sizes and types of models, demonstrating its consistent advantage over (Q)LoRA.

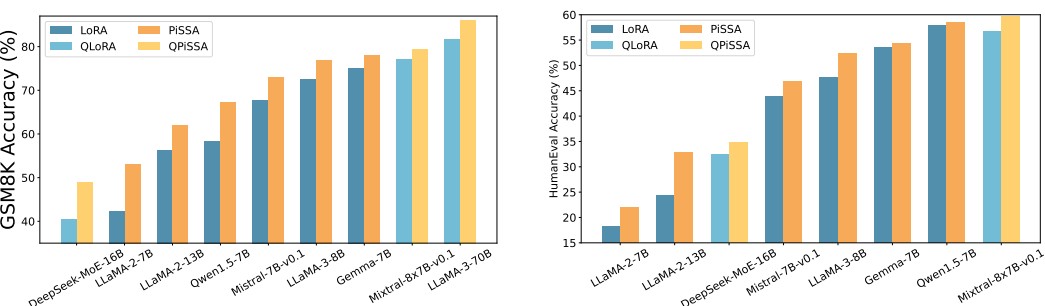

Figure 6: Comparison of (Q)PiSSA and (Q)LoRA across models from 7B to 70B.

## 5.5 Experiments on Various Ranks

This section explores the impact of incrementally increasing the rank of PiSSA/QPiSSA and LoRA/QLoRA from 1 to 128, aiming to determine whether PiSSA/QPiSSA consistently outperforms LoRA/QLoRA under different ranks. The training is conducted using the MetaMathQA-100K dataset for 1 epoch, while the validation is performed on the GSM8K and MATH datasets. The outcomes of these experiments are depicted in Figure 7, with additional results presented in Appendix K.

Figure 7a illustrates the quantization error reduction ratio across various ranks. In this figure, QLoRA shows no reduction in quantization error, while QPiSSA consistently outperforms LoftQ in reducing quantization error across all ranks, with a particularly notable advantage at lower ranks. In Figure 7b, the final loss on the training set is shown for models trained with ranks ranging from 1 to 128. The results indicate that PiSSA and QPiSSA achieve a better fit to the training data compared to LoRA, QLoRA, and LoftQ. In Figures 7c and Figures 7d, we compare the accuracy of the fine-tuned models on the GSM8K and MATH validation sets under various ranks, finding that PiSSA consistently outperforms LoRA with the same amount of trainable parameters. Furthermore, as the rank increases, PiSSA will reach and surpass the performance of full-parameter fine-tuning.

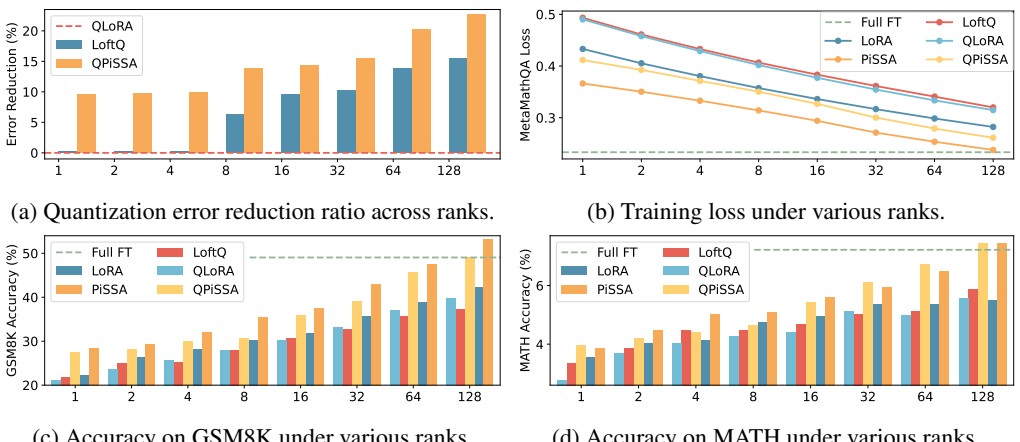

(a) Quantization error reduction ratio across ranks.

(b) Training loss under various ranks.

(c) Accuracy on GSM8K under various ranks.

(d) Accuracy on MATH under various ranks.

Figure 7: The comparison among (Q)LoRA, (Q)PiSSA, LoftQ, and full fine-tuning across ranks.

# 6 Conclusion

This paper presents a PEFT technique that applies singular value decomposition (SVD) to the weight matrix of pre-trained models. The principal components obtained from the SVD are used to initialize a low-rank adapter named PiSSA, while the residual components are kept frozen, to achieve effective fine-tuning and parameter efficiency simultaneously. Through extensive experiments, we found that PiSSA and its 4-bit quantization version QPiSSA significantly outperform LoRA and QLoRA in both NLG and NLU tasks, across different training steps, various model sizes and types, and under various amount of trainable parameters. PiSSA provides a novel direction for research in PEFT by identifying and fine-tuning the principal components within the model, analogous to *slicing and re-baking the richest slice of a pizza*. As PiSSA shares the same architecture as LoRA, it can be seamlessly used in existing LoRA pipelines as an efficient alternative initialization method.

# 7 Limitation

There are still some questions with PiSSA not addressed in this paper: 1) Besides language models, can PiSSA also be adapted to convolutional layers and enhance the performance of vision tasks? 2) Can PiSSA also benefit from some improvements to LoRA, such as AdaLoRA [58] and DyLoRA [66] which adaptively adjust the rank? 3) Can we provide more theoretical explanations for the advantages of PiSSA over LoRA? We are actively exploring these questions. Nevertheless, we are excited to see the huge potential of PiSSA already demonstrated in existing experiments and look forward to more tests and suggestions from the community.

# 8 Acknowledgements:

This work is supported by the National Key R&D Program of China (2022ZD016030).

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

# The Supplementary Material for The Paper "PiSSA: Principal Singular Values and Singular Vectors Adaptation of Large Language Models."

- In Section A, we combined PiSSA with two improved LoRA methods, and the experimental results show that these improvements can further enhance the effectiveness of PiSSA.

- In Section B, we use fast singular value decomposition to initialize PiSSA. The results indicate that the performance of fast singular value decomposition approaches that of SVD decomposition in just several seconds. This ensures that the cost of converting from LoRA/QLoRA to PiSSA/QPiSSA is negligible.

- In Section C, we demonstrate that the trained PiSSA adapter can be losslessly converted to LoRA, allowing for integration with the original model, facilitating sharing, and enabling the use of multiple PiSSA adapters.

- In Section D, we explore the experimental effects of using different precisions.

- In Section E, we discuss the effects of QPiSSA during multiple rounds of SVD decomposition, which can significantly reduce quantization errors without increasing training or inference costs.

- In Section F, we compare the use of high, medium, and low singular values and vectors to initialize adapters. The experimental results show that initializing adapters with principal singular values and vectors yields the best fine-tuning performance.

- In Section G, we used a normal distribution function to fit all linear layers of multiple models and calculated their mu and sigma. The experimental results show that after using PiSSA for initialization, the distribution of the remaining models, as described in Section 3 of the main text, is indeed narrower than that of the original models.

- In Section H, we provide a comprehensive comparison of quantization errors among QLoRA, LoftQ, and QPiSSA, theoretically explaining why QPiSSA reduce quantization errors.

- In Section I, we combine QPiSSA with various quantization methods beyond Normal Float 4bit, including INT8 and GPTQ. QPiSSA effectively reduces quantization error in these formats, enhancing fine-tuning performance.

- In Section J, we trained Mistral-7B and Gemma-7B for a sufficient number of steps. The results indicate that PiSSA and LoRA are less prone to overfitting compared to full parameter fine-tuning.

- In Section K, we offer a more detailed comparison of PiSSA and LoRA at different ranks. It is evident that PiSSA consistently outperforms LoRA in terms of loss convergence, quantization error reduction, and final performance across different ranks.

- In Section L, we describe the detail setting for NLU task.

# A Enhancing PiSSA with LoRA Improvement Methods

AdaLoRA introduces three improvements over LoRA:

- Trainable parameters in AdaLoRA are changed to $A, B$, and $E$. $A$ and $B$ are Gaussian-initialized, and $E$ is a zero-initialized $r$-dimensional vector, making $Adiag(E)B = \Delta W$, similar to singular value decomposition.
- A regularization loss $|AA^T - I| + |B^T B - I|$ is used to make $A$ and $B$ gradually orthogonal during training, resembling the SVD of $\Delta W$.
- An initial large rank is set, and less important E values are gradually masked during training, resulting in different final ranks for each layer, achieving better performance with the same number of parameters.

Despite the extensive use of SVD terms, AdaLoRA **does not perform actual SVD on any matrix**. In the PEFT domain, terms like low-rank decomposition, and singular value decomposition often appear. They generally refer to products of low-dimensional matrices approximating an ideal $\Delta W$ without actual matrix decomposition. To our knowledge, PiSSA is the first to perform SVD on the original model, fine-tuning the principal component while keeping the residual model frozen.

PiSSA and AdaLoRA represent different improvements to LoRA, making them combinable. Therefore, we additionally improved PiSSA based on AdaLoRA's three innovations:

- After extracting the principal singular values and vectors of $W$, we use $S$ as an independent trainable vector instead of multiplying it into $U$ and $V$.
- Since PiSSA's $U$ and $V$ are orthogonal at the beginning, maintaining their orthogonality through orthogonal regularization is very easy.
- Although AdaLoRA claims to dynamically reduce the number of trainable parameters, the initially large number of parameters is not truly pruned, resulting in more parameters being updated during actual training. Therefore, we did not use this improvement.

DoRA adds a learnable magnitude module to LoRA, normalizing $W + AB$ at each update step and multiplying its by the magnitude module. This allows $A, B$ to learn the direction and the magnitude module to learn the magnitude of $\Delta W$. While this approach can improve fine-tuning performance, normalizing $W + AB$ at each step results in slower fine-tuning speeds. In contrast, PiSSA only changes LoRA's initialization method, matching LoRA in training speed and converging faster, thereby reducing training costs.

Table 5: GSM8K accuracy for LoRA and PiSSA when combined with LoRA improvement methods.

| Model | Method | LoRA+ | PiSSA+ |
|---|---|---|---|
| | Vanilla | 71.01±0.199 | **76.75±0.036** |
| LLaMA-3-8B | DoRA | 72.38±0.189 | **77.51±0.257** |
| | AdaLoRA | 72.31±0.202 | **78.59±0.199** |

PiSSA, with its intrinsic principal singular values and orthogonal singular vectors, is very suitable for combination with AdaLoRA. According to Table 5. The performance of the improved PiSSA surpasses all the other methods including PiSSA. From lines 1, and 2 of the table, it is evident that the performance of PiSSA combined with DoRA significantly surpasses that of DoRA alone and also exceeds the performance of PiSSA alone. Taking into account the combination experiments of PiSSA with AdaLoRA, it can be inferred that PiSSA benefits from the enhancement techniques of LoRA, demonstrating the potential of PiSSA when integrated with other methods.

# B Fast Singular Value Decomposition

In order to speed up the decomposition of the pre-trained matrix $W$, we adopted the algorithm proposed by Halko et.al [47] (denoted as Fast SVD), which introduces randomness to achieve an approximate matrix decomposition. We compare the initialization time, error, and training loss between SVD and Fast SVD, with the results shown in Table 6. Initialization time refers to the computation time taken to decompose the pre-trained parameter matrix $W$, measured in seconds. Initialization error indicates the magnitude of the discrepancy introduced by Fast SVD compared to SVD after decomposing the matrix. Specifically, the error is the sum of the absolute differences between the matrices decomposed by original SVD and Fast SVD. For the error, we report the results of the self-attention module in the table. Loss refers to the loss value at the end of training. In Fast SVD, the parameter niter refers to the number of subspace iterations to conduct. A larger niter leads to increased decomposition time but results in smaller decomposition error. The symbol $\infty$ represents the experimental results with the SVD method.

Table 6: Comparation between SVD and Fast SVD in terms of initialization time, error and training loss.

| Metric | Niter | Rank | | | | | | | |
|---|---|---|---|---|---|---|---|---|---|
| | | 1 | 2 | 4 | 8 | 16 | 32 | 64 | 128 |
| Initialize Time | 1 | 5.05 | 8.75 | 5.07 | 8.42 | 5.55 | 8.47 | 6.80 | 11.89 |
| | 2 | 4.38 | 4.71 | 4.79 | 4.84 | 5.06 | 5.79 | 7.70 | 16.75 |
| | 4 | 5.16 | 4.73 | 5.09 | 5.16 | 5.60 | 7.01 | 7.90 | 11.41 |
| | 8 | 4.72 | 5.11 | 5.14 | 5.40 | 5.94 | 7.80 | 10.09 | 14.81 |
| | 16 | 6.24 | 6.57 | 6.80 | 7.04 | 7.66 | 9.99 | 14.59 | 22.67 |
| | $\infty$ | 434.92 | 434.15 | 434.30 | 435.42 | 435.25 | 437.22 | 434.48 | 435.84 |
| Initialize Error | 1 | 1.30E-3 | 1.33E-3 | 1.55E-3 | 1.9E-3 | 1.98E-3 | 1.97E-3 | 2.00E-3 | 1.93E-3 |
| | 2 | 5.84E-4 | 1.25E-3 | 1.45E-3 | 1.43E-3 | 1.48E-3 | 1.55E-3 | 1.48E-3 | 1.33E-3 |
| | 4 | 6.01E-4 | 8.75E-4 | 6.75E-4 | 1.10E-3 | 1.05E-3 | 1.03E-3 | 1.08E-3 | 9.75E-4 |
| | 8 | 1.26E-4 | 2.34E-4 | 5.25E-4 | 7.25E-4 | 5.75E-4 | 8.25E-4 | 8.25E-4 | 7.75E-4 |
| | 16 | 7.93E-5 | 2.25E-4 | 1.28E-4 | 6.50E-4 | 4.25E-4 | 6.50E-4 | 6.00E-4 | 4.75E-4 |
| | $\infty$ | – | – | – | – | – | – | – | – |
| Training Loss | 1 | 0.3629 | 0.3420 | 0.3237 | 0.3044 | 0.2855 | 0.2657 | 0.2468 | 0.2301 |
| | 2 | 0.3467 | 0.3337 | 0.3172 | 0.2984 | 0.2795 | 0.2610 | 0.2435 | 0.2282 |
| | 4 | 0.3445 | 0.3294 | 0.3134 | 0.2958 | 0.2761 | 0.2581 | 0.2414 | 0.2271 |
| | 8 | 0.3425 | 0.3279 | 0.3122 | 0.2950 | **0.2753** | 0.2571 | 0.2406 | 0.2267 |
| | 16 | 0.3413 | 0.3275 | **0.3116** | 0.2946 | 0.2762 | 0.2565 | 0.2405 | 0.2266 |
| | $\infty$ | **0.3412** | **0.3269** | **0.3116** | **0.2945** | 0.2762 | **0.2564** | **0.2403** | **0.2264** |

It can be observed that the computation time of the SVD is tens of times that of Fast SVD. In addition, SVD exhibits consistently high time consumption with minimal variation as the rank increases, while Fast SVD, although experiencing a slight increase in computation time with higher ranks, remains significantly lower compared to SVD throughout. As the rank increases, the initialization error initially rises gradually, with a slight decrease observed when the rank reaches 128. And at the same rank, increasing the niter in Fast SVD leads to a gradual reduction in error. For training loss, we observed that as the rank increases, the training loss decreases gradually. At the same rank, with the increase of niter, the training loss of models initialized based on Fast SVD approaches that of models initialized based on SVD.

## C Equivalently Converting PiSSA into LoRA

The advantage of PiSSA lies in its ability to significantly enhance training outcomes during the fine-tuning phase. After training, it allows for the direct sharing of the trained matrices $A$ and $B$. However, if we directly save $A, B$, users need to perform singular value decomposition on the original model to get $W^{res}$, which requires additional time. When employing fast singular value decomposition, there can be slight inaccuracies too. More importantly, such a way necessitates altering the parameters of the original model, which can be inconvenient when using multiple adapters, especially when some adapters might be disabled or activated. Therefore, we recommend converting the trained PiSSA module equivalently into a LoRA module, thereby eliminating the need to modify the original model's parameters during sharing and usage. In the initialization phase, PiSSA decomposes the original matrix into principal components and a residual matrix: $W = W^{res} + AB$. Upon completion of training, the model adjusts the weights as follows: $W + \Delta W = W^{res} + A'B'$. Thus, the modification of the model weights by PiSSA is given by:

$$\Delta W = A'B' - AB \tag{9}$$

$$= \underbrace{[A' \ A]}_{\Delta A} \underbrace{\begin{bmatrix} B' \\ -B \end{bmatrix}}_{\Delta B} \tag{10}$$

where $\Delta A \in \mathbb{R}^{m \times 2r}$ and $\Delta B \in \mathbb{R}^{2r \times n}$. Therefore, we can store and share the new adaptor $\Delta A$ and $\Delta B$ instead of $A', B'$, which allows directly inserting the adaptor to the original matrix and avoids breaking $W$. Since $r$ is typically small, the twice storage overhead is still acceptable. This modification allows for plug-and-play usage without the need for singular value decomposition, saving time and avoiding computational errors associated with the SVD, without necessitating changes to the original model parameters.

## D Comparison of Fine-Tuning in BF16 and FP32 Precision

In this section, we compare the effects of training with BFloat16 and Float32 precision. The comparing include four models: LLaMA-2-7B, Mistral-7B, Gemma-7B, and LLaMA-3-8B, each fine-tuned with all parameters in both BFloat16 and Float32 precision on the MetaMathQA-395K dataset. The validation results conducted on the GSM8K dataset are shown in Figure 7.

Table 7: Comparison of fine-tuning results of LLaMA-2-7B, Mistral-7B, Gemma-7B, and LLaMA-3-8B in BF16 and FP32 precision on MetaMathQA-395K dataset for 3 epochs.

| Model | Training Loss | | GSM8K ACC (%) | | MATH ACC (%) | |
|---|---|---|---|---|---|---|
| | BF16 | FP32 | BF16 | FP32 | BF16 | FP32 |
| LLaMA-2-7B | 0.1532 | **0.1316** | 63.15 | **68.31** | 13.14 | **20.38** |
| Mistral-7B | **0.1145** | 0.1306 | **73.09** | 65.88 | **26.44** | 23.66 |
| Gemma-7B | **0.1331** | 0.1382 | 75.21 | **75.97** | **29.18** | 28.64 |
| LLaMA-3-8B | **0.1271** | 0.1317 | **81.96** | 75.44 | **33.16** | 28.72 |

From Table 7, it is evident that the choice of precision greatly affects the experimental results. For example, the LLaMA-2-7B model shows a 5.16% higher performance on the GSM8K dataset when using FP32 compared to BF16. Conversely, the Mistral-7B and LLaMA-3-8B on GSM8K are 7.21% and 6.52% lower with FP32 than with BF16 separately. The Gemma-7B model shows similar performance with both precisions. Unfortunately, the experiments did not prove which precision is better. To reduce training costs, we use BF16 precision when fine-tuning all parameters. For methods with lower training costs, such as LoRA, PiSSA, we use FP32 precision. For QLoRA, QPiSSA and LoftQ, the base model was used NF4 precision, while the adapter layers used FP32 precision.

# E    Reducing Quantization Error through Multiple Iteration of SVD

Table 8 provides a supplementary explanation of the results in Table 4. When number of iterations $T > 1$, LoftQ uses an $N$-bit quantized weight $Q \in \mathbb{R}_N^{m \times n}$ and low-rank approximations $A \in \mathbb{R}^{m \times r}$ and $B \in \mathbb{R}^{n \times r}$ to minimize the following objective by alternating between quantization and singular value decomposition:

$$\min_{Q,A,B} \|W - (Q + AB^\top)\|_F, \tag{11}$$

where $\|\cdot\|_F$ denotes the Frobenius norm, $A$ and $B$ are set to zero. Inspired by LoftQ, our QPiSSA $T$-iter alternately minimize the following objective:

$$\min_{W_{res},A,B} \|W - (nf4(W_{res}) + AB^\top)\|_F, \tag{12}$$

where $A$ and $B$ are initialized by the principal singular values and singular vectors. The process is summarized in Algorithm 1:

---

**Algorithm 1** QPiSSA-$T$-iters, $T \geq 2$

---

**input** Pre-trained weight $W$, target rank $r$, 4-bit quantization function $nf4(\cdot)$, alternating step $T$
 1: Initialize $A_0, B_0 \leftarrow \text{SVD}(W)$ by (2) and (3)
 2: Initialize residual weight $W_{res} \leftarrow W - A_0 B_0^\top$
 3: **for** t = 2 to $T$ **do**
 4:     Update $A_t, B_t \leftarrow \text{SVD}(W - nf4(W_{res}))$ by (2) and (3)
 5:     Update residual weight $W_{res} \leftarrow W - A_{t-1} B_{t-1}^\top$
 6: **end for**
**output** $nf4(W_{res}), A_T, B_T$

---

Table 8: PiSSA reduces more quantization error on various ranks and number of iterations.

|  | Method | Rank | niter | Q | K | V | O | Gate | Up | Down | AVG |
|---|---|---|---|---|---|---|---|---|---|---|---|
| LLaMA -2-7B | QLoRA | – | – | 0 | 0 | 0 | 0 | 0 | 0 | 0 | 0 |
|  | loftQ | 128 | 1 | 8.1 | 8.1 | 7.2 | 7.3 | 5.3 | 5.1 | 5.1 | 6.6 |
|  | **PiSSA** | **128** | **1** | **19.0** | **18.1** | **8.9** | **8.9** | **8.2** | **5.9** | **6.0** | **10.7** |
|  | loftQ | 128 | 5 | 16.5 | 16.5 | 15.9 | 16.0 | 12.4 | 12.4 | 12.3 | 14.6 |
|  | **PiSSA** | **128** | **5** | **27.9** | **27.2** | **18.7** | **18.6** | **15.8** | **13.6** | **13.6** | **19.4** |
| LLaMA -3-8B | QLoRA | – | – | 0 | 0 | 0 | 0 | 0 | 0 | 0 | 0 |
|  | LoftQ | 64 | 1 | 4.3 | 11.0 | 9.9 | 3.9 | 2.7 | 2.5 | 2.6 | 5.3 |
|  | **PiSSA** | **64** | **1** | **11.3** | **16.4** | **8.8** | **6.3** | **4.5** | **2.9** | **3.3** | **7.7** |
|  | loftQ | 64 | 5 | 10.1 | 18.8 | 18.2 | 9.9 | 7.1 | 7.1 | 7.1 | 11.2 |
|  | **PiSSA** | **64** | **5** | **17.1** | **27.3** | **19.5** | **12.1** | **8.9** | **7.2** | **7.6** | **14.3** |
|  | loftQ | 128 | 1 | 8.2 | 20.7 | 18.8 | 7.5 | 5.2 | 4.8 | 4.9 | 10.0 |
|  | **PiSSA** | **128** | **1** | **17.1** | **26.5** | **10.7** | **10.7** | **7.0** | **5.0** | **5.6** | **11.8** |
|  | loftQ | 128 | 5 | 16.4 | 29.8 | 28.8 | 16.1 | 11.9 | 11.7 | 11.7 | 18.1 |
|  | **PiSSA** | **128** | **5** | **26.3** | **41.7** | **32.3** | **20.1** | **14.4** | **12.5** | **12.9** | **22.9** |
| LLaMA -3-70B | QLoRA | – | – | 0 | 0 | 0 | 0 | 0 | 0 | 0 | 0 |
|  | LoftQ | 64 | 1 | 2.4 | 11.6 | 9.2 | 1.9 | 1.8 | 1.7 | 1.3 | 4.3 |
|  | **PiSSA** | **64** | **1** | **12.3** | **25.0** | **9.0** | **4.1** | **4.2** | **3.2** | **2.2** | **8.6** |
|  | LoftQ | 64 | 5 | 6.1 | 17.8 | 17.0 | 6.0 | 4.3 | 4.4 | 4.2 | 8.5 |
|  | **PiSSA** | **64** | **5** | **15.7** | **34.2** | **18.9** | **7.5** | **6.7** | **5.7** | **4.7** | **13.4** |
|  | **PiSSA** | **128** | **1** | **17.7** | **36.6** | **15.7** | **6.7** | **5.8** | **4.5** | **3.8** | **13.0** |
|  | **PiSSA** | **128** | **5** | **23.2** | **49.0** | **30.5** | **12.5** | **10.1** | **8.8** | **8.2** | **20.3** |

According to Table 8, multiple iterations can significantly reduce quantization error. For instance, using QPiSSA-r64 with 5-iter on LLaMA-3-8B reduces the quantization error nearly twice as much as with 1-iter. In the main paper, we used 5 iterations in Section 5.3 and Section 5.4, while 1 iteration was used in Section 5.5.

# F  Conductive Experiments on Various SVD Components

To investigate the influence of singular values and vectors of varying magnitudes on the fine-tuning performance, we initialize the adapters injected into LLaMA 2-7B, Mistral-7B-v0.1, and Gemma-7B with principal, medium, and minor singular values and vectors. These models are then fine-tuned on the MetaMathQA dataset [2] and evaluated against the GSM8K [54] and MATH datasets [67], with the outcomes depicted in Figures 8.

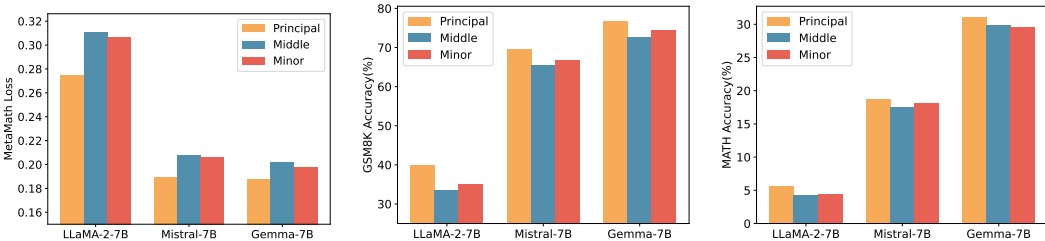

Figure 8: Initializing with principal, medium, and minor singular values and vectors, the training loss on the MetaMathQA and the accuracy on the GSM8K and MATH validation sets are reported, respectively, for three models.

The results highlight that initializing adapters with principal singular values and vectors consistently leads to reduced training loss and enhanced accuracy on both the GSM8K and MATH validation datasets across all three models. This underscores the efficacy of our strategy in fine-tuning the model parameters based on the principal singular values.

# G  The Residual Matrices having a Narrower Distribution

To intuitively compare the distribution differences between quantized original and residual models, in Figure 3, we took LLaMA 2-7B's first Query layer as an example to illustrate the distribution of $W$ and $W_{res}$. However, using only one layer of one model is not statistically significant. In this section, we applied PiSSA initialization to LLaMA-2-7B, Mistral-7B, Gemma-7B, and LLaMA-3-8B, and fit the values in every linear layer with Gaussian distribution and calculated their mu and sigma.

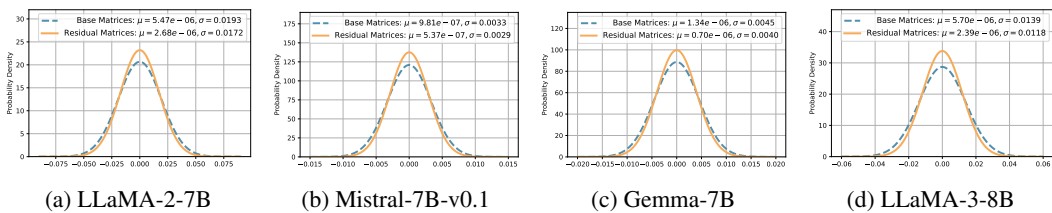

(a) LLaMA-2-7B      (b) Mistral-7B-v0.1      (c) Gemma-7B      (d) LLaMA-3-8B

Figure 9: Comparison of Loss and Ratio to the target A and target B for LoRA and PiSSA across the initial 5 steps.

The results in Figure 9 show that the residual models' means are closer to 0, and the standard deviations are smaller after PiSSA initialization. Thus, $W^{res}$ indeed has a narrower distribution than W in a statistical sense. Nevertheless, the difference is not as large as that in the first layer after averaging all layers, which we suspect is because middle layers in a model tend to have more even eigenvalue distributions due to redundancy and insufficient training.

## H    Comparing the Quantization Error of QLoRA, LoftQ and QPiSSA

This section extends the discussion in Section 4 by providing a comprehensive comparison of the quantization errors associated with QLoRA, LoftQ, and QPiSSA. Using the "layers[0].self_attn.q_proj" of LLaMA 2-7B as an example, we illustrate the singular values of critical matrices during the quantization process with QLoRA, LoftQ, and PiSSA in Figure 10. A larger sum of the singular values (nuclear norm) of the error matrix indicates a greater quantization error.

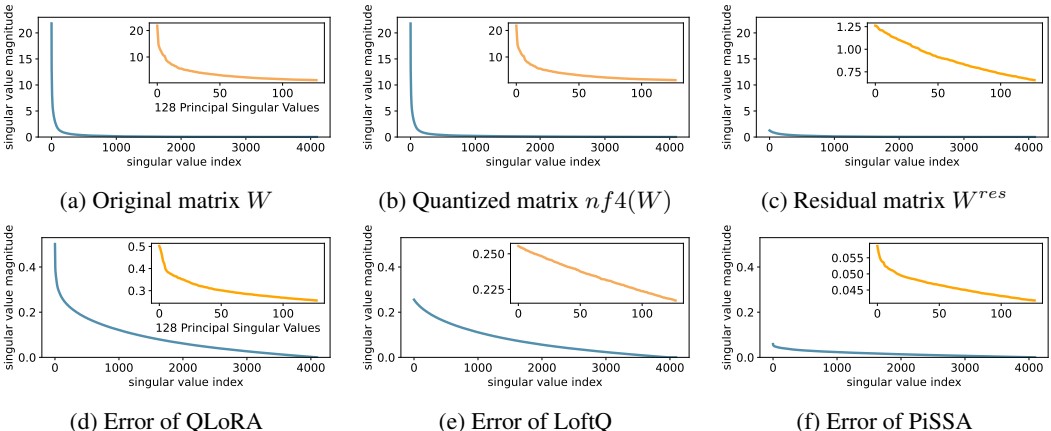

Figure 10: Several important singular values for calculation the quantization error of QLoRA, LoftQ and PiSSA.

The quantization error of QLoRA, which quantizes the base model to Normal Float 4-bit (NF4) and initializes $A$ and $B$ with Gaussian-Zero initialization, is:

$$\text{Quantization Error of QLoRA} = ||W - (nf4(W) + AB)||_* = ||W - nf4(W)||_*, \quad (13)$$

As shown in Equation 13, QLoRA decomposes the original matrix in Figure 10a into the sum of a quantized matrix (Figure 10b) and an error matrix (Figure 10d). By comparing Figure 10a and Figure 10d, we can see that the magnitude of the error matrix is much smaller than that of the original matrix. Therefore, the benefit of preserving the principal components of the $W$ matrix with the adapter is greater than that of preserving the principal components of the error matrix with the adapter.

LoftQ [14], designed to preserve the principal components of the error matrix using the adapter, first performs singular value decomposition on the quantization error matrix of QLoRA:

$$U^{err} S^{err} V^{err} = W - nf4(W), \quad (14)$$

then uses the larger singular values to initialize $A$ and $B$, thereby reducing the quantization error to:

$$LoftQ^{err} = ||W - (nf4(W) + AB)||* = ||U^{err}_{[r:]} S^{err}_{[r:,r:]} V^{err}_{[r:]}||* = \sum_{i=r}^{min(m,n)} S^{err}_{[i,i]}. \quad (15)$$

LoftQ eliminates only the largest $r$ singular values $S^{\text{err}}_{[:r]}$ (see Figure 10e) from the QLoRA error matrix (Figure 10d).

Our PiSSA, however, **does not quantify the base model but the residual model**:

$$\text{Quantization Error of PiSSA} = ||W - (nf4(W^{res}) + AB)||_* = ||W^{res} - nf4(W^{res})||_*, \quad (16)$$

where $A$ and $B$ are initialized following Equation 2 and 3. Since the residual model has removed the large-singular-value components, the value distribution of $W^{res}$ can be better fitted by a Student's t-distribution with higher degrees of freedom compared to $W$ (as can be seen in Figure 11) and thus quantizing $W^{res}$ results in lower error using 4-bit NormalFloat (shown in Figure 10f).

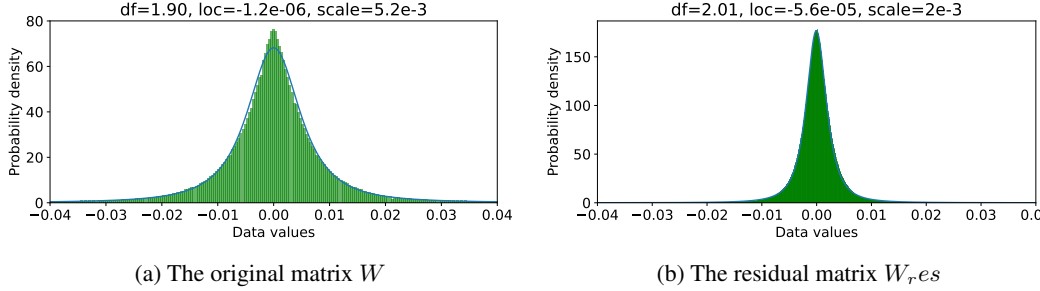

(a) The original matrix $W$           (b) The residual matrix $W_res$

Figure 11: Fitting the original matrix and the residual matrix using Student's t-distribution.

# I  Combining QPiSSA with Various Quantization Methods

In addition to NF4 quantization, QPiSSA can also be combined with GPTQ and INT8 quantization. We posit that PiSSA effectively reduces quantization error for several reasons:

- It reduces outlier values;
- It makes the value distribution more Gaussian-like;
- It preserves larger values in full precision, thereby narrowing the weight distribution in the quantized portion.

While INT8 also targets the reduction of outlier values (point 1), PiSSA has the potential to enhance this effect. The second point aligns well with NF4, and the third point is crucial as PiSSA uses an adaptor to retain a significant portion of weights in full precision, maintaining the integrity of critical values.

Table 9: Quantization Error and Accuracy for PiSSA Combined with Various Quantization Methods. GPTQ quantizes each row $w$ independently, adjusting one weight at a time while updating all remaining, non-quantized weights. Therefore, the nuclear norm method used for calculating quantization error in the main paper is not applicable to GPTQ. Instead, we measure Perplexity on WikiText-2, where a lower Perplexity indicates reduced quantization error.

| Model | Dataset | Quantization Error | | GSM8K Accuracy | |
|---|---|---|---|---|---|
| | | QLoRA | PiSSA | QLoRA | PiSSA |
| LLaMA-3-8B | NF4 | 324.8 (nuclear norm) | **265.8** (nuclear norm) | 70.79±0.42 | **73.76±0.20** |
| | INT8 | 34.47 (nuclear norm) | **28.21** (nuclear norm) | 71.68±0.14 | **76.54±0.32** |
| | GPTQ | 20.79 (perplexity) | **6.23** (perplexity) | 70.18±0.42 | **74.58±0.22** |

As shown in Table 9, QPiSSA combined with INT8 reduces quantization error by **18.16%** on LLaMA-3-8B, and significantly outperforms QLoRA using INT8. Furthermore, in row 3 of the table, the perplexity of LLaMA-3-8B increases to **20.79** after quantization with GPTQ-4bit on the C4 dataset. However, when PiSSA is applied, the perplexity is reduced to **6.23**. These results confirm the effectiveness of PiSSA in reducing quantization error, as discussed in the main paper.

Overall, QPiSSA demonstrates a clear advantage over QLoRA when combined with various quantization methods, retaining the fast convergence and superior performance characteristics of PiSSA while minimizing quantization error.

# J  Evaluating PiSSA on Mixtral and Gemma with More Training Steps

This is the supplement for Section 5.2. We applied PiSSA, LoRA, and full parameter fine-tuning on the full MetaMathQA-395K dataset using Mistral-7B and Gemma-7B models, training for 3 epochs. Figures 12 and 13 display the training loss, gradient norm, and evaluation accuracy on GSM8K.

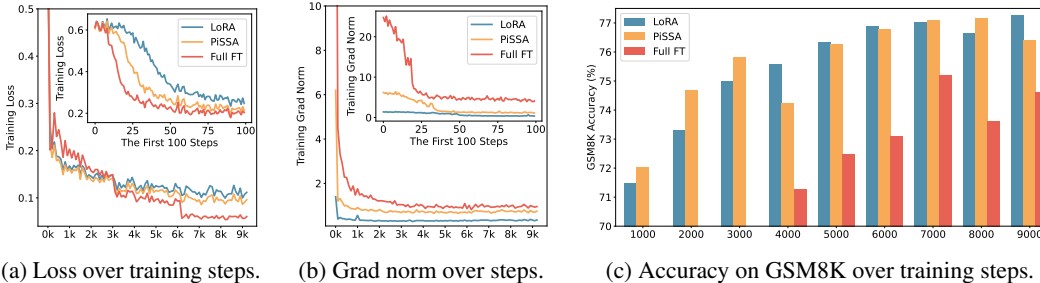

(a) Loss over training steps.  (b) Grad norm over steps.  (c) Accuracy on GSM8K over training steps.

Figure 12: Fine-tuning Mistral-7B-v0.1 on the MetaMathQA-395K dataset for 3 epochs: A comparison of full parameter fine-tuning (indicated by a dashed line), LoRA (in blue), and PiSSA (in orange).

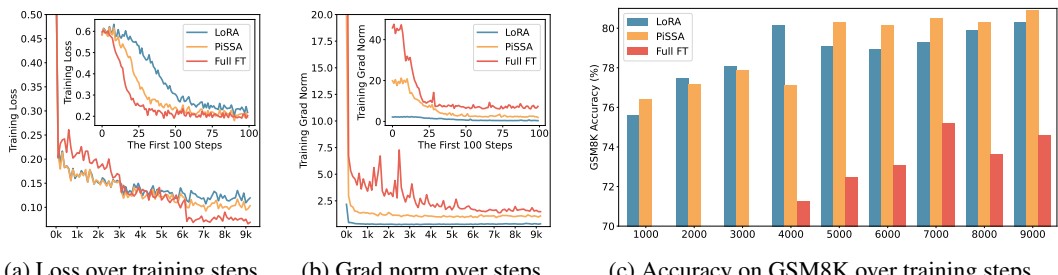

(a) Loss over training steps.  (b) Grad norm over steps.  (c) Accuracy on GSM8K over training steps.

Figure 13: Fine-tuning Gemma-7B on the MetaMathQA-395K dataset for 3 epochs: A comparison of full parameter fine-tuning (indicated by a dashed line), LoRA (in blue), and PiSSA (in orange).

As shown in Figure 12a and 13a, the loss for full parameter fine-tuning decreases sharply with each epoch, indicating overfitting to the training data. Notably, during the entire first epoch, the loss for full parameter fine-tuning on Mistral and Gemma is significantly higher than for LoRA and PiSSA, suggesting that full parameter fine-tuning has weaker generalization capabilities compared to LoRA and PiSSA on Mistral-7B and Gemma-7B models. The gradient norm for the first epoch in Figure 13b fluctuates dramatically with each step, further indicating instability in the training process for full parameter fine-tuning. Consequently, as illustrated in Figures 12c and 13c, the performance of full parameter fine-tuning is markedly inferior to that of LoRA and PiSSA. These experiments demonstrate that using parameter-efficient fine-tuning can prevent the over-fitting issue caused by over-parameters.

## K   Supplementary Experiments on Various Ranks

### K.1   Quantization Error for More Type of Layers

Figure 7a only shows the reduction ratio of quantization error for "q_proj" layers. In Figure 14, we present the error reduction ratios for the remaining types of linear layers under different ranks.

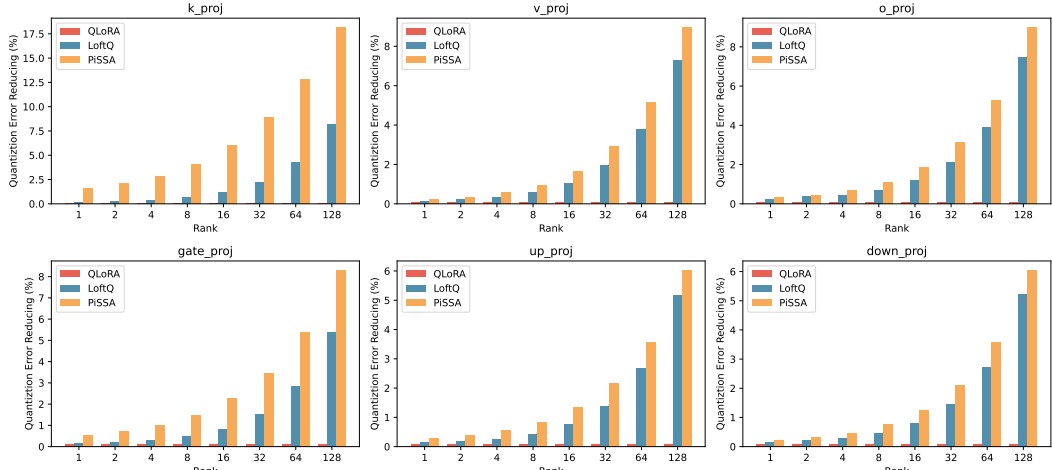

Figure 14: Comparison of quantization errors in QLoRA, LoftQ, and PiSSA across k_proj, v_proj, o_proj and gate_proj, up_proj, down_proj layers.

From Figure 14 it can be observed that under different ranks, the reduction ratio of quantization error for various linear layers in LLaMA-2-7B, including "k_proj", "v_proj", "o_proj", "gate_proj", "up_proj", and "down_proj" layers, is consistently lower with PiSSA compared to LotfQ.

## K.2 Evaluation Performance for More Model on Various Ranks

Section 5.5 only validated the effectiveness of LLaMA-2-7B. In Figure 15, we also present the comparative results of Mistral-7B-v0.1, and Gemma-7B under different ranks.

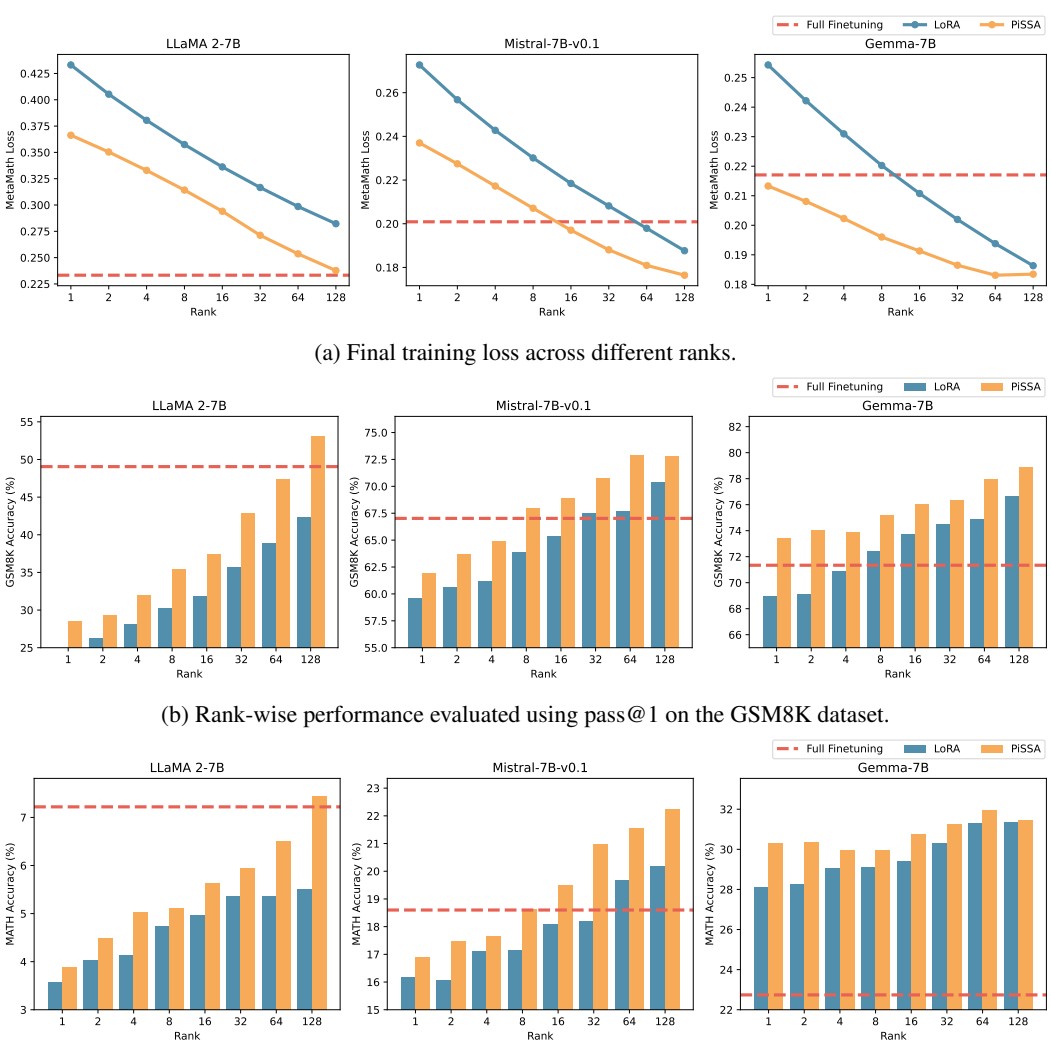

(a) Final training loss across different ranks.

(b) Rank-wise performance evaluated using pass@1 on the GSM8K dataset.

(c) Rank-wise performance evaluated using pass@1 on the MATH dataset.

Figure 15: Fine-tuning LLaMA 2-7B, Mistral-7B-v0.1, and Gemma-7B on the MetaMathQA dataset: A comparison of full parameter fine-tuning (indicated by a dashed line), LoRA (in blue), and PiSSA (in orange).

From Figure 15, PiSSA uses fewer trainable parameters compared to LoRA while achieving or even surpassing full-parameter fine-tuning on LLaMA-2-7B and Mistral-7B. Remarkably, on Gemma-7B, PiSSA exceeds full-parameter fine-tuning performance even at rank=1. However, as the rank increases to 128, the performance of PiSSA begins to decline, indicating that PiSSA over-parameterizes earlier than LoRA. This over-parameterization phenomenon does not occur on LLaMA-2-7B, suggesting that increasing the rank further might enable PiSSA to achieve even higher performance on LLaMA-2-7B.

## K.3 More Training Loss and Grad Norm under Various Ranks

In Figure 16 and 17, we examining the loss and gradient norm during the training process of PiSSA and LoRA on LLaMA 2-7B, Mistral-7B-v0.1, and Gemma-7B using different ranks.

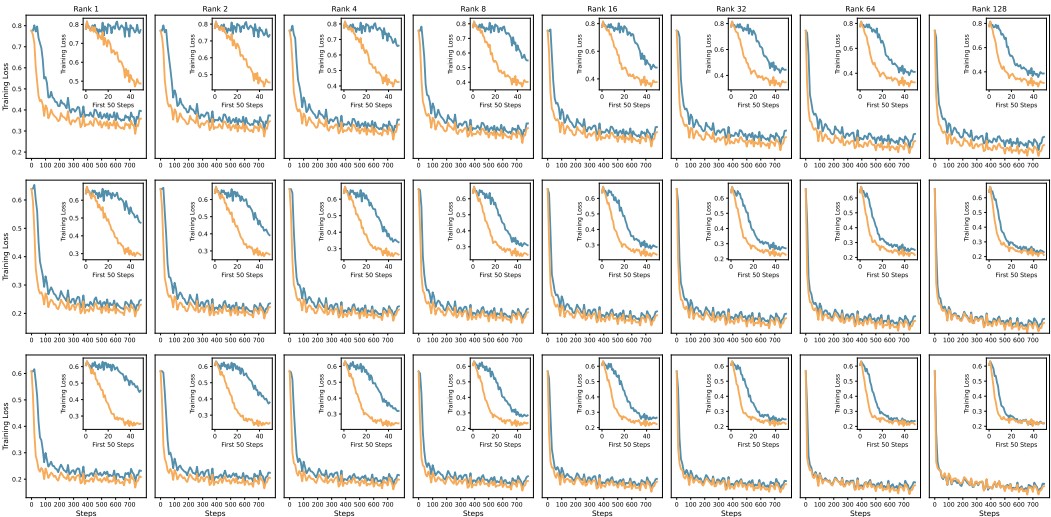

Figure 16: Comparison of training loss for LLaMA-2-7B, Mistral-7B, and Gemma-7B, organized into three rows, using LoRA and PISSA across ranks $2^i, i \in [0, 7]$, organized into eight columns.

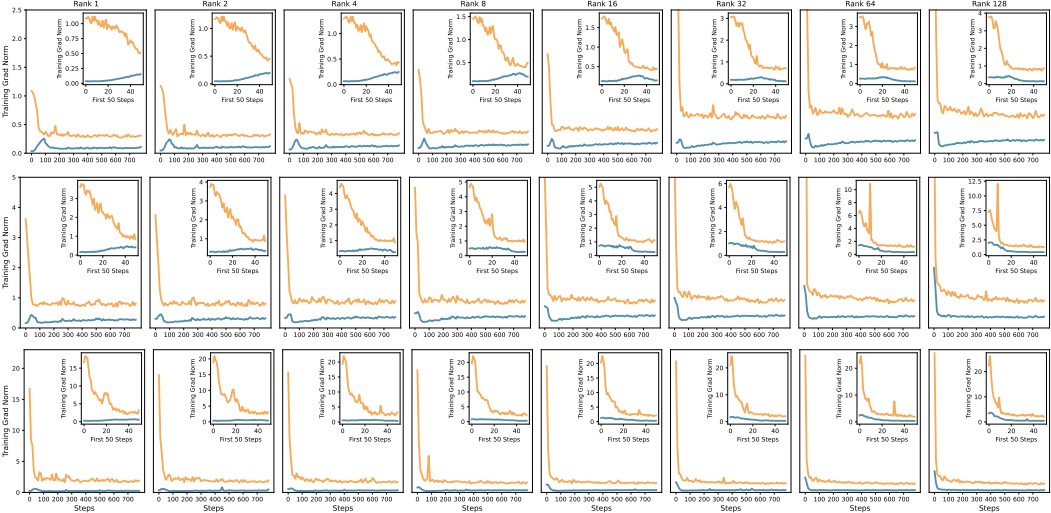

Figure 17: Comparison of grad norm for LLaMA-2-7B, Mistral-7B, and Gemma-7B, organized into three rows, using LoRA and PISSA across ranks $2^i, i \in [0, 7]$, organized into eight columns.

From Figure 16, PiSSA consistently shows a faster initial loss reduction compared to LoRA across various ranks. Additionally, the final loss remains lower than that of LoRA. This advantage is particularly pronounced when the rank is smaller. From Figure 17, the gradient norm of PiSSA remains consistently higher than that of LoRA throughout the training process, indicating its efficient fitting of the training data. A closer look at the first few steps of LoRA's gradient norm reveals a trend of rising and then falling. According to Section 3, LoRA's gradients are initially close to zero, leading to very slow model updates. This requires several steps to elevate LoRA's weights to a higher level before subsequent updates. This phenomenon validates our assertion that LoRA wastes some training steps and therefore converges more slowly. It demonstrates the robustness of the faster convergence property of PiSSA across various ranks.

# L   Experimental Settings on NLU

**Datasets**   We evaluate the performance of PiSSA on GLUE benchmark, including 2 single-sentence classification tasks (CoLA, SST), 5 pairwise text classification tasks (MNLI, RTE, QQP, MRPC and QNLI) and 1 text similarity prediction task (STS-B). We report overall matched and mismatched accuracy on MNLI, Matthew's correlation on CoLA, Pearson correlation on STS-B, and accuracy on the other datasets.

**Implementation Details**   To evaluate the performance of PiSSA intuitively, we compared PiSSA to LoRA with the same number of trainable parameters. DeBERTa-v3-base has $184M$ trainable parameters. PiSSA and LoRA were applied to $W_Q$, $W_K$ and $W_V$ respectively, resulting in a total of $1.33M$ trainable parameters.

The results for full fine-tune, BitFit [15], HAdapter [30], PAdapter [36], LoRA with Gassian initialization [11] and AdaLoRA are sourced from AdaLoRA [58], based on five runs. The remaining results use the publicly available LoftQ [14] codebase and are averaged over three runs. In LoRA, the B matrix is initialized to zero, while the A matrix can be initialized using various methods, such as Gaussian initialization and Kaiming initialization [68]. The selection of the initialization method can influence the final results. In this paper, we report the different results of LoRA based on Gaussian initialization and Kaiming initialization in the experiments, as shown in Table 2 and Table 3. For DoRA, we used the code from the PEFT package for deployment and conducted a search on key hyperparameters. We set the rank of PiSSA in this experiment as 8 and selecte lora alpha in 8, 16. We utilize AdamW with linear learning rate schedule to optimize and tune learning rate (LR) from 1e-4,2e-4,3e-4,4e-4,5e-4, 6e-4, 5e-5, 3e-5. Batch sizes (BS) are selected from 6, 8, 16, 32. The hyperparameter configurations of PiSSA, DoRA and LoRA with Kaiming Initialization are shown in Table 10. LoRA$^K$ denotes LoRA with Kaiming initialization, and $\alpha$ denotes LoRA alpha.

Table 10: Hyperparameters of PiSSA, DoRA and LoRA with Kaiming Initialization on GLUE.

| Dataset | PiSSA | | | | DoRA | | | | LoRA$^K$ | | | |
|---|---|---|---|---|---|---|---|---|---|---|---|---|
| | Epoch | BS | LR | $\alpha$ | Epoch | BS | LR | $\alpha$ | Epoch | BS | LR | $\alpha$ |
| MNLI | 5 | 16 | 5e-4 | 8 | 10 | 32 | 2e-4 | 16 | 10 | 32 | 3e-4 | 8 |
| SST-2 | 20 | 16 | 3e-5 | 8 | 10 | 16 | 4e-4 | 16 | 10 | 32 | 1e-4 | 8 |
| MRPC | 20 | 32 | 2e-4 | 8 | 10 | 32 | 4e-4 | 16 | 10 | 32 | 4e-4 | 8 |
| CoLA | 20 | 16 | 1e-4 | 8 | 20 | 8 | 1e-4 | 6 | 30 | 32 | 4e-4 | 8 |
| QNLI | 10 | 32 | 1e-4 | 16 | 10 | 16 | 2e-4 | 16 | 25 | 32 | 3e-4 | 8 |
| QQP | 10 | 16 | 1e-4 | 8 | 10 | 16 | 1e-4 | 6 | 10 | 16 | 3e-4 | 8 |
| RTE | 50 | 16 | 1e-4 | 8 | 50 | 8 | 2e-4 | 6 | 50 | 32 | 4e-4 | 8 |
| STS-B | 20 | 8 | 3e-4 | 8 | 20 | 16 | 3e-4 | 6 | 30 | 16 | 4e-4 | 8 |

# M    Comparison of Initial Gradient Subspaces

To compare the gradient subspaces of PiSSA and LoRA, we conducted two additional experiments to validate our analysis.

First, we trained LLaMA-3-8B on the MetaMath dataset five times, initializing LoRA with different random seeds while using the same batch of 128 training examples to compute LoRA's gradients. After performing dimensionality reduction to two dimensions, the results are presented in Table 11.

Table 11: Ablation results for LoRA and PiSSA across different seeds.

|        | Method | Seed 0 | Seed 1 | Seed 2 | Seed 3 | Seed 4 |
|--------|--------|--------|--------|--------|--------|--------|
| grad_A | LoRA   | [0,0]  | [0,0]  | [0,0]  | [0,0]  | [0,0]  |
|        | PiSSA  | **[0,1]** | **[0,1]** | **[0,1]** | **[0,1]** | **[0,1]** |
| grad_B | LoRA   | [-0.99, 0.12] | [0.95, 0.31] | [0.46, -0.89] | [0.24, 0.97] | [0.04, -0.99] |
|        | PiSSA  | **[1,0]** | **[1,0]** | **[1,0]** | **[1,0]** | **[1,0]** |

We observe that the gradient of matrix $A$ remains consistently zero, while the gradient direction of matrix $B$ varies across initializations. This behavior arises because matrix $A$'s gradient depends on matrix $B$, which in LoRA is initialized to zero, resulting in a zero gradient for $A$. In contrast, matrix $B$ is initialized from a Gaussian distribution, leading to variation in its gradient direction across different seeds. In comparison, PiSSA's gradient direction remains consistent across all five training runs, as it solely depends on the original model and the training data. This experiment highlights the stability of PiSSA's optimization trajectory relative to LoRA's more variable directionality.

Next, we quantitatively compared the effect of updating along the principal singular value direction versus a "random" direction during the early stages of fine-tuning. We trained LLaMA-3-8B on the MetaMathQA dataset using both PiSSA and LoRA, saving the parameters and gradients from the first 50 iterations. At the 50th step, the loss values for LoRA and PiSSA were 0.3677 and 0.2899, respectively. Using the parameters from the 50th step as the target point, we evaluated the movement in the first five steps relative to the target, computing how much progress was made towards the final point. We then divided this progress by the total target distance to obtain a ratio. These ratios are shown in Figure 18.

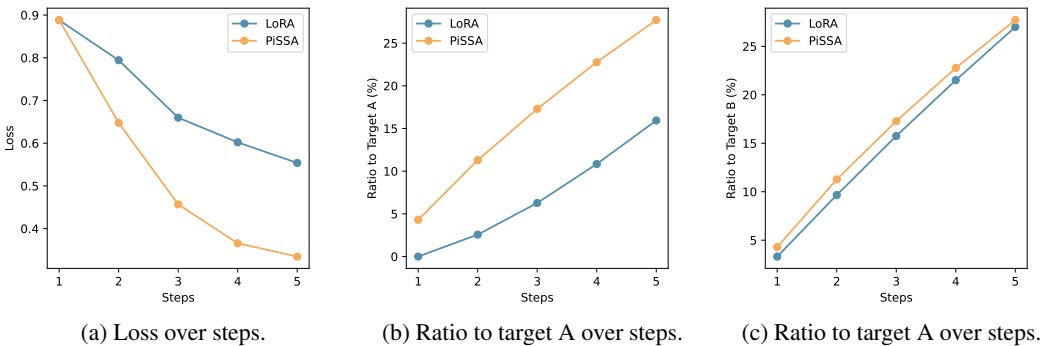

(a) Loss over steps.        (b) Ratio to target A over steps.        (c) Ratio to target A over steps.

Figure 18: Comparison of Loss and Ratio to the target A and target B for LoRA and PiSSA across the initial 5 steps.

The results reveal that after just five updates, PiSSA reduced the loss from 0.8884 to 0.3346, while LoRA's loss reduction was more modest, dropping to only 0.5538. This demonstrates the advantage of updating along the principal singular value direction, which PiSSA leverages, leading to faster convergence. Further, in the first step, matrix $A$ in LoRA exhibited a zero gradient and therefore did not update. Over the next four steps, it moved only 15.94% towards the target direction. Similarly, matrix $B$ in LoRA consistently moved less towards the target endpoint compared to PiSSA.

