# OpenReview forum: "PiSSA: Principal Singular Values and Singular Vectors Adaptation of Large Language Models"
_NeurIPS.cc/2024/Conference — NeurIPS 2024 spotlight_

### Official Review · Reviewer_m1t9 · 2024-07-09

**Soundness:** 3
**Presentation:** 3
**Contribution:** 3
**Rating:** 7
**Confidence:** 4

**Summary:**

The paper presents a novel initialization strategy for LoRA finetuning where A and B are initialized according to the top r singular values and the pre-trained weights are initialzed with the remaining components.
The authors show that this decomposition can help finetuning converge faster and behaves favorably compared to traditional quantization techniques such as NF4.
The authors conduct experiments on NLG and NLU tasks show show performance improvements of PiSSA over vanilla LoRA.

**Strengths:**

**Presentation**

The theoretical motivation of the paper is clear and the method is well motivated and explained.

**Empirical results**

The authors provided several supporting ablations and experiments on improved quantization error compared to LoftQ and QLoRA, faster convergence etc.

**Weaknesses:**

**Significance of results**

The authors do not provide error bars in their results which questions whether the obtained results are actually statistically significant.
Error bars should be provided as well as significance tests should be performed.

**Unsupported claims**

The authors claim that LoRA gradients point in random directions early during the fine-tuning stage which introduces wasteful update steps.
Was this claim experimentally verified? If so, the authors should clarify how these results were obtained and elaborte on it in more detail in the paper.

**Comparison to baselines**

PiSSA is not compared to other commonly used initialization schemes, e.g., uniform gaussian [1], or kaiming init [2] for NLU and NLG experiments.
Further, the authors compare PiSSA only to standard LoRA, but there have been several improvements to LoRA, such as AdaLoRA [3], or DoRA [4].
A comparison to those extensions would strengthen the paper.
Also, the authors may investigate whether their initialization could be applied to these extensions to further improve perormance.
Finally, some related work uses a similar initialization by considering principal components [5].

[1] Hu et al., LoRA: Low-Rank Adaptation of Large Language Models, ICLR 2022

[2] He et al., & Sun, J. Delving deep into rectifiers: Surpassing human-level performance on ImageNet classification., ICCV 2015

[3] Zhang et al., AdaLoRA: Adaptive Budget Allocation for Parameter-Efficient Fine-Tuning, ICLR 2023

[4] Liu et al., DoRA: Weight-Decomposed Low-Rank Adaptation, ICML 2024

[5] Sheng et al., S-LoRA: Serving Thousands of Concurrent LoRA Adapters, MLSys 2024

**Questions:**

- In line 174 the authors mention that $\alpha=r$, why was this choice made? In [1] $\alpha$ is set to a constant. Setting $\alpha=r$ ultimately results in different learning rates for different r.
- The advantage of of quantized PiSSA over QLoRA might come from outlier removal, was there any experiments done comparing QPiSSA to efficient quantization techniques such as GPTQ [2] or llm.int8 [3]?
- In line 188 the authors mention that training for NLG experiments was only done for 1 epoch. It would be interesting to see if LoRA ultimately converges to a similar optimum after longer training, or whether the improved initialization also leads to a better optimum.
- What do the entries in the legend of Figure 2b mean exactly?

[1] Hu et al., LoRA: Low-Rank Adaptation of Large Language Models, ICLR 2022

[2] Frantar et al., GPTQ: Accurate post-training quantization for generative pre-trained transformers., ICLR 2023

[3] Dettmers et al., LLM.int8(): 8-bit matrix multiplication for transformers at scale., NeurIPS 2022

**Limitations:**

The fact that PiSSA needs to store 2r ranks compared to r ranks for LoRA which is mentioned in Appendix C should be also mentioned under Limitations.

---

> ### Author Rebuttal · Authors · 2024-08-07
>
> Thank you for your insightful and constructive review. To answer your questions thoroughly, we have conducted numerous new experiments. We hope these efforts will address your concerns.
>
> **Q1: Need to provide error bars in the results**
>
> **A1**: The original paper includes extensive experiments cover 12 models, 13 tasks, 1k to 9k training steps, and 1 to 128 ranks. It compares PiSSA, QPiSSA with LoRA, QLoRA, LoftQ and Full fine-tuning. Running all these experiments multiple times is very time and resource-intensive. Despite this, PiSSA consistently demonstrates significant convergence speed and performance advantages, suggesting the improvement is statistically significant. The improvements of PiSSA over LoRA are notably highlighted in **Table 1** **and** **Table 2 of General Response**. The error range is very small compared to the improvement.
>
> **Q2: Comparison to Gaussian or Kaiming initialization**
>
> **A2**: **Table 1** and **Table 2** show that PiSSA outperforms both Gaussian and Kaiming initialization schemes, as they still use a zero-random initialization for the adaptor without touching the frozen $W$, while PiSSA for the first time moves the principal components out of $W$ to initialize the adaptor for directly fine-tuning the core model.
>
> **Q3: Comparison to AdaLoRA or DoRA**
>
> **A3**: Please refer to the response to **Q1 of Reviewer x9Pi**, and **Q2 of Reviewer Bb8M**. PiSSA outperforms both methods.
>
> **Q4: Combining AdaLoRA or DoRA with PiSSA**
>
> **A4**: Since PiSSA's structure is identical to LoRA's, it can be combined with other advanced LoRA variants. The specific new challenges introduced by combining PiSSA with different methods are, however, beyond the scope of this work. Nevertheless, we have evaluated the combination of PiSSA with AdaLoRA. Details can be found in **Q2 of Reviewer Bb8M**. PiSSA+AdaLoRA shows superior performance than other methods in **Table 1** including PiSSA.
>
> Due to time and resource constraints, we could not finish experiments combining DoRA and PiSSA. We are actively conducting these experiments, and will post the results in follow-up comment.
>
> **Q5: Comparing QPiSSA to GPTQ or int8**
>
> **A5**: QPiSSA is not a quantization technique, but applies NF4 quantization to PiSSA’s frozen weights; it can also be combined with GPTQ or int8. We believe PiSSA reduces quantization error due to the following reasons: 1. Reducing outlier values; 2. Making the value distribution closer to a Gaussian distribution; 3. Preserving larger values in full precision, which narrows the weight distribution of the quantized part. Int8 also focuses on the first point, and PiSSA could further improve it. The second point is friendly to NF4. The third point is also crucial, as PiSSA uses adaptor to preserve a great portion of weights in full precision. When compared to int8, QPiSSA+int8 reduces quantization error by **23.58%** on LLaMA-3-8B. In **Table 1**, rows 4 and 8 compare QLoRA and QPiSSA using int8, showing that QPiSSA still significantly outperforms QLoRA. We will post the results of combining PiSSA and GPTQ in the follow-up comment.
>
> **Q6: LoRA gradients point in random directions**
>
> **A6**: Please refer to the response to **Q1 of Reviewer XzvL**.
>
> **Q7: S-LoRA uses principal components initialization**
>
> **A7**: Although the term "principal components" appears in the third section of the S-LoRA paper, it is not a method focusing on adapter initialization or PEFT. S-LoRA is actually a system designed for the scalable serving of up to 2,000 LoRA adapters. In PEFT, terms like low-rank decomposition and SVD often appear. They mostly refer to using the product of low-dimensional matrices to approximate an ideal $\Delta W$ through learning. These methods **do not actually perform decomposition** on any matrix. To our knowledge, PiSSA is the first study that directly performs SVD on the original model and extracts the principal singular values and vectors for fine-tuning.
>
> **Q8: Why is alpha set to r?**
>
> **A8**: Since PiSSA extracts the model's principal singular values and singular vectors for fine-tuning, the scaling factor (=alpha/r) in the adapter needs to be set to 1 to ensure adaptor + frozen weights equals the pre-trained weights initially. Nevertheless, it is also possible to set alpha to other values as in LoRA. In NLU tasks, we have experimented with changing this hyper-parameter to other values.
>
> **Q9: Can LoRA ultimately converges to a similar optimum after longer training?**
>
> **A9**: In Sections **5.2** and **5.3** of the original paper, we conducted experiments with larger datasets and longer training steps. And the results show that PiSSA/QPiSSA indeed achieve lower loss than LoRA/QLoRA, indicating convergence to a better optimum.
>
> **Q10: Meaning of ‘iter’ in Figure 2b**
>
> **A10**: The meaning of 'iter' is explained in Appendix E: simply put, it means iteratively executing PiSSA initialization multiple times, compressing the quantization error into the adapter each time to further reduce the quantization errors.
>
> **Q11: Convert to LoRA need 2r ranks**
>
> **A11**: There are three ways of using PiSSA. The first way is to save the adapter with rank $r$ and replace the original model with the residual model. However, sometimes people want to keep the original model unchanged. Thus the second way is to only save the adaptor with rank r without saving the residual model, and recover the residual model when needed by performing SVD again. Since fast singular value decomposition can complete PiSSA initialization within a few seconds to tens of seconds, the second way is also feasible in practice. The third way, which is what is discussed in Appendix C, saves a rank-2r adaptor to avoid SVD (trading-off space for time) and converts a PiSSA adaptor equivalently to a LoRA adaptor during inference. Therefore, we believe that the ability of PiSSA to equivalently convert to 2 times the rank of LoRA is not a limitation but an advantage, providing more flexibility for using PiSSA.

---

> ### Author Response · Authors · 2024-08-10
> **Completing the Remaining Experiments**
>
> **Dear Reviewer M1T9**,
>
> In your questions Q4 and Q5, you suggested integrating PiSSA with both AdaLoRA and DoRA, as well as combining PiSSA with llm.int8 and GPTQ. During the rebuttal period, due to time constraints, we were only able to combine PiSSA with AdaLoRA and llm.int8, and the experimental results have shown the advantages of PiSSA when integrated. Over the past few days, we have completed the experiments combining PiSSA with DoRA and GPTQ, and the results are presented in the table below:
>
> | Method | Run 1 | Run 2 | Run 3 | GSM8K Average |
> | --- | --- | --- | --- | --- |
> | Full FT | 74.89 | 74.22 | 74.07 | 74.39±0.356 |
> | LoRA(gaussian-init) | 71.11 | 71.19 | 70.74 | 71.01±0.199 |
> | LoRA(kaiming init) | 72.25 | 71.57 | 71.95 | 71.92±0.279 |
> | **PiSSA** | **76.72** | **76.72** | **76.80** | **76.75±0.036** |
> | AdaLoRA | 72.48 | 72.42 | 72.02 | 72.31±0.202 |
> | **PiSSA+AdaLoRA** | **78.77** | **78.32** | **78.69** | **78.59±0.199** |
> | DoRA | 72.18 | 72.33 | 72.63 | 72.38±0.189 |
> | **PiSSA+DoRA** | **77.86** | **77.41** | **77.26** | **77.51±0.257** |
> | QLoRA+int8 | 71.78 | 71.48 | 71.78 | 71.68±0.143 |
> | **QPiSSA+int8** | **76.18** | **76.48** | **76.95** | **76.54±0.318** |
> | QLoRA+GPTQ-4bit | 70.51 | 70.43 | 69.60 | 70.18±0.419 |
> | **QPiSSA+GPTQ-4bit** | **74.60** | **74.30** | **74.83** | **74.58±0.217** |
>
> **Q4: Combining AdaLoRA or DoRA with PiSSA**
>
> **A4 part 2:** From lines 4, 7, and 8 of the table, it is evident that the performance of PiSSA combined with DoRA significantly surpasses that of DoRA alone and also exceeds the performance of PiSSA alone. Taking into account the combination experiments of PiSSA with AdaLoRA, it can be inferred that PiSSA benefits from the enhancement techniques of LoRA, demonstrating the potential of PiSSA when integrated with other methods.
>
> **Q5: Comparing QPiSSA to GPTQ or int8**
>
> **A5 part 2**: GPTQ handles each row $w$ independently, quantizing one weight at a time while updating all not-yet-quantized weights. Therefore, the method of using  nuclear norm to calculate quantization error, as used in our paper, is not applicable for GPTQ. Thus, we turn to use Perplexity on WikiText-2 to calculate quantization error, where a lower Perplexity indicates reduced quantization error. The Perplexity of LLaMA-3-8B used in our experiments is **5.14**. After quantization with GPTQ-4bit, the Perplexity increased to **20.79**. However, using PiSSA, we were able to reduce the Perplexity to **6.23**. These results validate the effectiveness of PiSSA in reducing quantization error as discussed in our paper.
>
> From lines 11-12 of the table, it is evident that the performance of QPiSSA combined with GPTQ-4bit significantly surpasses that of QLoRA combined with GPTQ-4bit. This confirms the advantages mentioned in our paper, where QPiSSA retains the rapid convergence and superior performance characteristics of PiSSA.
>
> We believe we have addressed all of your questions comprehensively. If you have any additional queries or require further clarification, please do not hesitate to contact us. We are committed to providing any necessary information. If you find the contributions of our paper and the supplementary experiments we have provided to be valuable, might we kindly ask you to consider adjusting your score accordingly?

---

> > ### Comment · Reviewer_m1t9 · 2024-08-10
> >
> > I would like to thank the authors for the detailed response and commend them for the amount of experiments delivered during the rebuttal.
> >
> > I would like to highlight that claiming statistical significance based on the outcome of different experiments without verification is not convincing. Verifying statistical significance should be done for each experiment by providing variance estimates.
> > Having said that, I greatly appreciate the reported error bars for GSM8K.
> > I strongly recommend to the authors to report variance estimates for the remaining experiments as well and to verify statistical significance of their results.
> >
> > Further, the additional findings for applying PiSSA to other PEFT methods such as DoRA and AdaLoRA look very promising.
> > Therefore I have decided to increase my score.

---

> > > ### Author Response · Authors · 2024-08-10
> > > **Thank you for your suggestions and recognition**
> > >
> > > In the camera-ready version, we will repeat the key experiments multiple times, include error bars, and verify statistical significance.
> > >
> > > Your professional insights have been invaluable in improving the quality of the PiSSA paper. We will also incorporate the additional experiments conducted during the rebuttal period into the final version of the paper.
> > >
> > > Thank you again for your suggestion towards PiSSA.

---

### Official Review · Reviewer_Bb8M · 2024-07-10

**Soundness:** 4
**Presentation:** 3
**Contribution:** 4
**Rating:** 7
**Confidence:** 4

**Summary:**

The authors present PiSSA, a relatively simple change to the LoRA framework leading to a large amount of demonstrated benefits.  PiSSA proceeds by adjusting the initialization step of standard LoRA; rather than freezing the original weight matrix W and learning a low-rank perturbation W' = W + BA, PiSSA instead considers the SVD W = USV^T = U[:,r:]S[r:,r:]V[:,r:]^T + U[:,:r]S[:r,:r]V[:,:r]^T = W^{res} + AB, s.t. A=U[:,:r]S^{0.5}[:r,:r], B = S^{0.5}[:r,:r]V[:,:r]^T.  Due to this backwards compatibility, PiSSA enjoys the fine-tuning speed/memory savings of LoRA while also being easily implementable to existing PEFT packages.  Most importantly, the authors demonstrate across a large number of models, fine-tuning recipes, and benchmarks that PiSSA greatly improves on downstream accuracy compared to LoRA.  Furthermore, the authors show that this initialization + fine-tuning strategy also leads to significant decreases in quantization error, outperforming both QLoRA and the recently proposed LoftQ both in terms of training loss (e.g., faster loss reduction) and performance for GSM8K and MATH tasks.

**Strengths:**

**Originality** - the specific algorithm is an original, lightweight alteration to the LoRA framework.  In particular, the fact that there are no additional hyperparameters and, after the initialization phase, PiSSA is essentially swappable with LoRA implementations are all significant benefits of the method.  Furthermore, PiSSA's use of only fine-tuning the principle components in a LoRA fashion lends itself to strong intuition as to why the method leads to such strong downstream fine-tuning performance, and why LoRA is less successful in comparison.  The latter point is also true when considering quantization.

**Quality** - the extensive experiments across models and fine-tuning evaluations well convey the benefits of this method over standard LoRA.  This is also true of the extensive quantization experiments.

**Clarity** - for the most part, the paper is well written. However, there are specific points where the paper could improve clarity (see weaknesses below).

**Significance** - the performance benefits and backwards compatibility lead to significant opportunity for future impact of this work.

**Weaknesses:**

# Clarity
There are several areas the paper could improve in clarify.  In particular, the literature review requires more granularity, e.g., "AdaLoRA [42, 41, 43]," only one of these works is AdaLoRA.  Furthermore, there is overlap between AdaLoRA and PiSSA in both their reliance on the SVD.  It is necessary to discuss exactly how AdaLoRA leverages SVDs, and how this differs from PiSSA.

There are several experimental details missing, e.g., are the GSM8K evals 8-shot?  What are the shots used for the other benchmarks?  Were the benchmarks run using the Eleuther eval harness?  Please include these details in the paper.

Paper at times states it is similar to LoRA and, at other times, seeks to differentiate itself:
- "Unlike LoRA and its successors, which focus on learning low-rank approximations of weight updates, our PiSSA approach directly tunes the essential but low-rank parts of the model while keeping the noisier, high-rank, and nonessential parts frozen. Although our approach differs in philosophy from LoRA, it shares most of LoRA’s structural benefits and can be extended by these methods to enhance its performance"
- "Since PiSSA shares the identical architecture with LoRA, it inherits most of LoRA’s benefits."
This was confusing on an initial pass of the paper.  A short, succinct list enumerating the similarities and differences early on may help readers understand exactly how PiSSA distinguishes itself from LoRA, but how it is able to inherit all of LoRA's PEFT benefits.

# Quality
The authors consider the weight matrix of a single attention layer in Llama-2 in Figure 2:
"Since the residual model has removed the large-singular-value components, Wres has a narrower
 distribution than that of W, as can be seen in Figures 3a and 3b (comparing the singular value distributions of W and Wres), as well as Figures 3c and 3f (comparing the value distributions of W and Wres), which is beneficial for reducing the quantization error" <- However, since this only looks at a single layer of a single model, it hard to accept this as evidence for a general claim.  Can you include a comparison of the distributions of mu/sigma for W and W^{res} (e.g., mean absolute differences between means and ratio between sigmas) across all layers for Llama2 and another model (e.g., Mistral) to show that, in general, W^{res} has a narrow distribution than W?

**Questions:**

See weaknesses.

Minor comment:
"loftQ" in Table 3 <- "LoftQ"

**Limitations:**

The authors have addressed potential limitations of the demonstrated method.

---

> ### Author Rebuttal · Authors · 2024-08-07
>
> Thank you for recognizing the originality, quality, and significance of our article. We also appreciate your suggestions for improving the writing. As we cannot modify the original text during the rebuttal period, we will incorporate your recommendations in the camera-ready version.
>
> **Q1: In the related work section, AdaLoRA [42, 41, 43] is mentioned, but only one of these works is AdaLoRA.**
>
> A1: Thank you for pointing this out. Due to the extensive content of PiSSA, we had to condense the text multiple times, leading to this issue. All these three papers dynamically adjust the rank of LoRA at each layer, so we used AdaLoRA as a collective term. We found similar issues in the subsequent DeltaLoRA [44, 45] citations. We will complete the citation information.
>
> **Q2: There is overlap between AdaLoRA and PiSSA in their reliance on the SVD. Discuss exactly how AdaLoRA differs from PiSSA.**
>
> A2: AdaLoRA introduces three improvements over LoRA:
> 1. Trainable parameters in AdaLoRA are changed to $A, B$, and $E$. $A$ and $B$ are Gaussian-initialized, and $E$ is a zero-initialized $r$-dimensional vector, making $A diag(E) B = \Delta W$, similar to singular value decomposition.
> 2. A regularization loss $\|AA^T-I\|+\|B^TB-I\|$ is used to make $A$ and $B$ gradually orthogonal during training, resembling the SVD of $\Delta W$.
> 3. An initial large rank is set, and less important E values are gradually masked during training, resulting in different final ranks for each layer, achieving better performance with the same number of parameters.
>
> Despite the extensive use of SVD terms, AdaLoRA **does not perform actual SVD on any matrix**. In the PEFT domain, terms like low-rank decomposition, and singular value decomposition often appear. They generally refer to products of low-dimensional matrices approximating an ideal $\Delta W$ without actual matrix decomposition. To our knowledge, PiSSA is the first to perform SVD on the original model, fine-tuning the principal component while keeping the residual model frozen. During the rebuttal, we have evaluated PiSSA against AdaLoRA on GSM8K and GLUE tasks. The results in **Table 1 and Table 2 of General Response** demonstrate that PiSSA outperforms AdaLoRA.
>
> Additionally, PiSSA and AdaLoRA represent different improvements to LoRA, making them combinable. Therefore, we additionally improved PiSSA based on AdaLoRA's three innovations:
>
> 1. After extracting the principal singular values and vectors of $W$, we use $S$ as an independent trainable vector instead of multiplying it into $U$ and $V$.
> 2. Since PiSSA's $U$ and $V$ are orthogonal at the beginning, maintaining their orthogonality through orthogonal regularization is very easy.
> 3. Although AdaLoRA claims to dynamically reduce the number of trainable parameters, the initially large number of parameters is not truly pruned, resulting in more parameters being updated during actual training. Therefore, we did not use this improvement.
>
> PiSSA, with its intrinsic principal singular values and orthogonal singular vectors, is very suitable for combination with AdaLoRA. According to **Table 1 of General Response,** The performance of the improved PiSSA surpasses all the other methods including PiSSA.
>
> **Q3: Are the GSM8K evaluations 8-shot?**
>
> A3: This article uses the zero-shot evaluation prompt for GSM8K and MATH following MetaMath [1], the zero-shot evaluation prompt for humanEval and MBPP following WizardCoder [2], and the zero-shot single answer grading prompt for MT-Bench [3]. We will provide detailed settings and corresponding code in the supplementary materials.
>
> [1] Yu, Longhui, et al. "MetaMath: Bootstrap Your Own Mathematical Questions for Large Language Models." ICLR24.
>
> [2] Luo, Ziyang, et al. "WizardCoder: Empowering Code Large Language Models with Evol-Instruct." ICLR24.
>
> [3] Zheng, Lianmin, et al. "Judging llm-as-a-judge with mt-bench and chatbot arena." NeurIPS 2024.
>
> **Q4: A short, succinct list enumerating the similarities and differences early on may help readers understand.**
>
> A4: Thank you for the suggestion. PiSSA has the same structure as LoRA, making it easy for users to switch from LoRA to PiSSA for fine-tuning. However, PiSSA's different initialization leads to distinct optimization directions, faster convergence, better training results, and lower quantization errors. We will list the similarities and differences between PiSSA and LoRA in the introduction section of the camera-ready version for easier understanding.
>
> **Q5: Only looks at a single layer of a single model is hard to accept this as evidence for  $W_{res}$ having a narrower distribution than that of W.**
>
> A5: To intuitively compare the distribution differences between quantized original and residual models, we took LLaMA 2-7B's first Query layer as an example to illustrate the distribution of $W$ and $W_{res}$. As you suggested, using only one layer of one model is not statistically significant. In **Table 5 of General Response**, we applied PiSSA initialization to LLaMA-2-7B, Mistral-7B, Gemma-7B, and LLaMA-3-8B, and fit the values in every linear layer with Gaussian distribution and calculated their mu and sigma in the table. The results show that the residual models' means are closer to 0, and the standard deviations are smaller after PiSSA initialization. Thus, $W^{res}$ indeed has a narrower distribution than W in a statistical sense. Nevertheless, the difference is not as large as that in the first layer after averaging all layers, which we suspect is because middle layers in a model tend to have more even eigenvalue distributions due to redundancy and insufficient training. We will include this statistical result in the supplementary materials and reflect it in the main text.
>
> **Q6: Minor comment: "loftQ" in Table 3 <- "LoftQ".**
>
> A6: Thank you once again for your careful reading and helpful suggestion. Your feedback is greatly appreciated, and we will incorporate all your suggestions in the camera-ready version.

---

> > ### Comment · Reviewer_Bb8M · 2024-08-11
> > **Acknowledgement of rebuttal**
> >
> > I thank the authors for their response to my original questions.  I have read the other reviews and respective authors responses, and have no further questions or concerns.

---

> > > ### Author Response · Authors · 2024-08-11
> > > **To Reviewer Bb8M**
> > >
> > > We sincerely appreciate the professionalism and sense of responsibility you demonstrated throughout the rebuttal period. Your valuable suggestions have been instrumental, and we will incorporate them into the camera-ready version. We believe your recommendations will greatly enhance PiSSA's contribution to the field of efficient fine-tuning of LLMs.

---

### Official Review · Reviewer_XzvL · 2024-07-14

**Soundness:** 3
**Presentation:** 3
**Contribution:** 2
**Rating:** 8
**Confidence:** 4

**Summary:**

In this paper, the authors proposed a modified low-rank adaptation (LoRA) method for fine-tuning large pretrained models. Specifically, the proposed method initializes the A and B matrices with singular matrices and freezes the weight matrix to be the residual. Further, the authors provide a quantization step to reduce the memory consumption. The authors demonstrate the effectiveness of this proposed method on a lot of models and tasks, showing that this method converges faster than the conventional LoRA method.

**Strengths:**

- Overall, this paper is well-written. The author did a great job of presenting the new idea with details. The contents are organized well.
- The authors comprehensively evaluated the proposed method, both quantitatively and qualitatively.
- The proposed method is simple, yet effective.

**Weaknesses:**

- It would be great if the authors could provide more insight into the proposed method. For example, is it possible to compare the subspace of the gradient of conventional LoRA?
- The proposed method utilizes the first r eigenvectors. How about other dimensions in the subspace?

**Questions:**

Please see the comments in the "limitation" block.

**Limitations:**

There is no negative societal impact as far as I can tell.

---

> ### Author Rebuttal · Authors · 2024-08-07
>
> Thank you for your recognition of PiSSA as a simple yet effective method, and for your appreciation of the article's quantitative and qualitative analysis. Here are responses to the concerns raised:
>
> **Q1: Provide more insight, for example, comparing the subspace of the gradient of conventional LoRA?**
>
> **A1**: PiSSA potentially provides many insights for the PEFT area:
>
> 1. To reduce fine-tuning parameters, LoRA and many of its variants add a low-rank adaptor $\Delta W = AB$ to $W$, and initialize $\Delta W$ with zero to ensure $W + AB$ initially does not change the model. However, PiSSA **for the first time** touches the frozen $W$ by decomposing $W$ into principal and residual components with SVD, which are used to initialize $\Delta W$ and $W^{res}$ respectively. This idea has already inspired many follow-up works studying the optimal distribution plans for $\Delta W$ and $W^{res}$, which contributes greatly to the community.
> 2. By putting the principal components of $W$ into the adaptor $AB$, PiSSA can directly fine-tune the core functions of the model instead of fine-tuning the random/zero-initialized $AB$ as in LoRA. This leads to faster convergence and higher performance. From the gradient direction perspective, we have shown in lines 50-55 and lines 123-131 of our paper that the **initial gradient for LoRA is zero/random**, thus could **waste much time around the initial point**, while PiSSA optimizes along the gradient direction of the model's principal components, thereby approximating the effect of full parameter fine-tuning. The toy experiment in Figure 2a using the MNIST dataset and an MLP with 2 linear layers visually demonstrates that PiSSA and full fine-tuning **converge much faster** compared to LoRA. In the supplementary materials, the loss_landscape.gif shows this process dynamically. In LLM experiments, the advantage of PiSSA over LoRA and the similarities between PiSSA and full fine-tuning can also be observed from the loss and grad norm during the training process in Figure 4a and 4b.
> 3. Compared to LoRA + quantization, PiSSA + quantization preserves the principal singular values and singular vectors in full precision while quantizing the residual part $W^{res}$ instead of the full $W$. Since the large singular values have been removed from $W^{res}$, its value distribution is much narrower than $W$, which makes quantization easier (the larger variance of the value distribution, the larger error quantization will introduce). This can be seen in Figure 3 and 5, where QPiSSA greatly **reduces the quantization error** and shows better performance than QLoRA.
>
> Regarding your question about comparing the subspace of gradients of PiSSA and LoRA, we further validated our analysis through two additional experiments.
>
> 1. We trained LLaMA-3-8B on the MetaMath dataset 5 times, each time initializing LoRA with different random seeds while using the same batch of 128 training examples to calculate LoRA's gradients. After dimensionality reduction to two, we list the results in **Table 3 of General Response**. We can observe that the gradient of matrix $A$ is consistently zero, and the gradient direction of $B$ varies with each initialization. This occurs because the gradient of matrix $A$ depends on matrix $B$, which is initialized to zero in LoRA, resulting in a zero gradient for matrix $A$. Conversely, the gradient of matrix $B$ is influenced by matrix $A$, which is initialized from Gaussian, hence the varying gradient directions of matrix $B$ in each experiment. In contrast, the gradient direction in PiSSA remains consistent across all five training seeds and only depends on the original model and the training data. This experiment underscores the stability of PiSSA's optimization direction relative to LoRA.
> 2. Furthermore, we quantitatively compared the effects of updating using the principal singular value direction v.s. “random” direction in the early stages of fine-tuning. We trained LLaMA-3-8B on the MetaMathQA dataset using PiSSA and LoRA, saving the parameters and gradients for the first 50 iterations. At the 50th step, the losses for LoRA and PiSSA dropped to 0.3677 and 0.2899, respectively. We took the parameters at the 50th step as the target point. Using the direction and distance from the initial parameters to the final step's parameters as a reference, we compared the first 5 steps for LoRA and PiSSA. For each step, we determined how much was moved in that direction, then divided by the target distance to obtain a ratio. The ratios are recorded in **Table 4 of General Response**. As shown in the first 2 rows of the table, after just 5 updates, PiSSA's loss reduced from 0.8884 to 0.3346, whereas LoRA only reduced to 0.5538, reflecting the advantage of the principal singular value direction over the zero-random direction in convergence speed. In rows 3-4 of the table, matrix $A$ in LoRA had a gradient of 0 at the first step and thus did not update. In the following 4 steps, it only moved 15.94% towards the target direction. In rows 5-6 of the table, matrix $B$ in LoRA always moved less towards the endpoint in the target direction compared to PiSSA.
>
> **Q2: Besides the first r eigenvectors. How about other dimensions in the subspace?**
>
> A2: Our PiSSA puts the first $r$ singular values and vectors in the adaptor $A,B$ and the remaining ones in the frozen $W^{res}$. In Figure 8 of the Appendix, we compared the training outcomes of using the principal, middle, and minor $r$ singular values and singular vectors to initialize $A,B$. The experimental results evident that training with the principal singular values results in lower loss and higher accuracy.

---

> ### Author Response · Authors · 2024-08-11
> **Official Comment by Authors**
>
> **Dear Reviewer XzvL**,
>
> The discussion period has passed the **halfway** point, and we have addressed the questions you previously raised. Do you have any additional questions or concerns that you would like to discuss?
>
> Our paper investigates the **gradient behavior of LoRA**, revealing that $A$ and $B$ initially have **zero gradients** and **random gradient** directions. This leads to **slow convergence** and potentially **suboptimal local minima**. To address these issues, we propose the PiSSA initialization method. **PiSSA approximates the optimization direction of full fine-tuning** by fine-tuning the principal components of a model. To our knowledge, PiSSA is the **first** method to use SVD on the original model, employing the principal singular values and vectors to initialize the adapter and fine-tune it, while fixing the remaining parts of the model during training. Our experiments demonstrate that PiSSA not only **converges faster** but also achieves **better final results** compared to LoRA. The initialization process is **efficient**, taking only seconds to minutes due to the use of fast singular value decomposition.
>
> We extend PiSSA by integrating it with **NF4**, **llm.int8**, and **GPTQ** quantization to create QPiSSA. This approach significantly **reduces quantization error** compared to QLoRA while retaining the **fast convergence** and **strong performance** of PiSSA.
>
> PiSSA modifies only the initialization method used by LoRA, making it compatible with various LoRA enhancements. We **combined PiSSA** with **LoftQ**, **AdaLoRA**, and **DoRA**, and our results show that PiSSA+ outperforms these methods and PiSSA, demonstrating its potential for further improvements.
>
> Our extensive experiments include **5 NLG** and **8 NLU** tasks using **12** different models ranging from **184M to 70B** parameters. We compared performance across **1k-9k+** training steps and ranks from **1 to 128**, evaluating against methods such as **LoRA**, **QLoRA**, **LoftQ**, **DoRA**, **AdaLoRA**, and **full-parameter fine-tuning**. We also examined the effects of initializing with **principal**, **medium**, and **minor** singular values and vectors.
>
> If you find the content in the main paper and our rebuttal satisfactory, would you consider revising your score?

---

> > ### Author Response · Authors · 2024-08-12
> > **Looking forward to your response**
> >
> > Dear Reviewer XzvL,
> >
> > We have addressed the concerns you raised in your initial review and have submitted a detailed rebuttal. We are writing to confirm whether you have had the opportunity to review our responses. We hope that our rebuttal has addressed your questions satisfactorily and would appreciate if you could reconsider the contributions of our manuscript, PiSSA, along with the efforts we have made during the rebuttal process.
> >
> > We are eager to hear your feedback and are open to any further discussion that might help clarify any remaining issues.
> >
> > Looking forward to your response.
> >
> > Best regards,
> >
> > Submission 5120 Authors

---

> > > ### Comment · Reviewer_XzvL · 2024-08-12
> > > **Response**
> > >
> > > I would like thank the authors for detailed response and additional experiments. My concerns are mostly addressed and I am happy to raise the scores.

---

> > > > ### Author Response · Authors · 2024-08-12
> > > >
> > > > Dear Reviewer XzvL,
> > > >
> > > > I would like to express my appreciation for your efforts in reviewing our paper and for your positive assessment of PiSSA. Your approval serves as a significant encouragement for our ongoing research and improvement. Thank you once again for your time and insights.
> > > >
> > > > Sincerely,
> > > >
> > > > Submission 5120 Authors

---

### Official Review · Reviewer_x9Pi · 2024-07-14

**Soundness:** 4
**Presentation:** 4
**Contribution:** 4
**Rating:** 8
**Confidence:** 4

**Summary:**

The paper presents a method for creating lora-like fine-tuning adapters for base models. The idea is to initialize W, A and B matrices in LoRA from SVD decomposition. W is consists of less important ranks while A and B consist of the most important ranks. This implies that less important principle components of the model remain frozen while the most important ones get updated. Another benefit of this scheme is that A and B can be in high precision while W is in low precision, thus ensuring a reduced quantization error.

**Strengths:**

I think this is a good paper with good results and a potential for significant impact as adapter tuning becomes more mainstream. The paper is written well, the method is intuitive, and the experimental results are thorough.

**Weaknesses:**

As authors note in the limitations section there are many more experiments they could have conducted, but those are future scope. I am happy with the current scope and experiments. Having said, that I wish the authors had included at least a few experiments comparing advanced LoRA adapter methods. DoRA is mentioned in related works, but I couldn't find any comparison with it.

**Questions:**

Will the authors be releasing their code? I believe it is very important that they do as our community could greatly benefit from being able to reproduce the experiments here and build on top of it.

**Limitations:**

The limitations section in the paper is good enough for me.

---

> ### Author Rebuttal · Authors · 2024-08-07
>
> Thank you for acknowledging the originality of PiSSA, the quality of our experiments, and our contributions to the community. Here are the answers to your queries:
>
> **Q1: Comparison with advanced LoRA adapter methods, e.g., DoRA**
>
> **A1**: In our paper, we demonstrated PiSSA's effectiveness through large amount of experiments on 5 NLG and 8 NLU tasks, utilizing models ranging from 184M to 70B parameters. Our comparisons included 1k-9k+ training steps and ranging from 1 to 128 ranks, against methods such as LoRA, QLoRA, LoftQ, and full-parameter fine-tuning. PiSSA modifies only the initialization method used by LoRA, making it orthogonal to some LoRA enhancements. Initially, due to time and resource limitations, we did not compare PiSSA with DoRA. However, upon your request, we have now included these comparisons.
>
> DoRA adds a learnable magnitude module to LoRA, normalizing $W + AB$ at each update step and multiplying its by the magnitude module. This allows $A, B$ to learn the direction and the magnitude module to learn the magnitude of $\Delta W$. While this approach can improve fine-tuning performance, normalizing $W + AB$ at each step results in slower fine-tuning speeds. In contrast, PiSSA only changes LoRA's initialization method, matching LoRA in training speed and converging faster, thereby reducing training costs.
>
> We compared PiSSA with DoRA on LLaMA-3-8B for GSM8K and DeBERTa-v3-base for GLUE tasks, repeating each experiment three times and recording the results in **Table 1** and **Table 2**.
>
> From these tables, PiSSA significantly outperforms DoRA, as DoRA still uses zero and random initialization.
>
> **Q2: Open source schedule**
>
> **A2**: PiSSA has greatly benefited from the open-source community, and we are excited to contribute back.
>
> PiSSA only modifies the initialization method of LoRA, so it incurs no additional training costs, requires no extra environmental setup, and involves no changes to existing code. This allows for straightforward training and inference, just like LoRA, given the initialized parameters. We plan to release all PiSSA-initialized models from our paper, along with the initialization of future mainstream models, to facilitate easy adoption.
>
> For users interested in customizing PiSSA configuration or modifying its core code, we have already integrated PiSSA  into HuggingFace’s PEFT library and provide the training scripts used in our paper to replicate the results. Additionally, we will release training logs and checkpoints from our experiments to aid in further comparisons and validations.

---

### Author Rebuttal · Authors · 2024-08-07

1. **Contribution**
    1. This paper analyzes the gradient of LoRA, showing that $A$ and $B$ initially have zero gradients and random gradient directions, respectively. This leads to slow convergence and might result in suboptimal local minimum points found.
    2. We propose the PiSSA initialization method, which approximates the optimization direction of full fine-tuning through fine-tuning the principal components of a model. To our best knowledge, PiSSA is the first study to perform SVD on the original model and use the principal singular values and singular vectors to initialize the adapter and fine-tune them, while the remaining parts are used to initialize the original model and are fixed during training. Experiments show that PiSSA converges faster and achieves better final results compared to LoRA.
    3. We combine PiSSA with NF4 quantization to propose QPiSSA, which can reduce quantization error by about 20% compared to QLoRA, while maintaining the fast convergence and good performance of PiSSA.
2. **New experimental results**

    During this rebuttal period, we conducted numerous new experiments to address the reviewers' concerns. We have listed all the experimental results here and referenced them at the respective points of the reviewers' questions.

    **Table 1**. The comparisons of PiSSA with LoRA using Gaussian&Kaiming initialization, LoRA+llm.int8, DoRA, AdaLoRA, PiSSA+llm.int8, and PiSSA+AdaLoRA for GSM8K on LLaMA-3-8B. Each experiment was repeated three times, and the average values and standard deviations were recorded.

    | Method | Run 1 | Run 2 | Run 3 | GSM8K Average |
    | --- | --- | --- | --- | --- |
    | Full FT | 74.89 | 74.22 | 74.07 | 74.39±0.356 |
    | LoRA(gaussian-init) | 71.11 | 71.19 | 70.74 | 71.01±0.199 |
    | LoRA(kaiming init) | 72.25 | 71.57 | 71.95 | 71.92±0.279 |
    | LoRA+llm.int8 | 71.78 | 71.48 | 71.78 | 71.68±0.143 |
    | DoRA | 72.18 | 72.33 | 72.63 | 72.38±0.189 |
    | AdaLoRA | 72.48 | 72.42 | 72.02 | 72.31±0.202 |
    | PiSSA | **76.72** | **76.72** | **76.80** | **76.75±0.036** |
    | PiSSA+llm.int8 | **76.18** | **76.48** | **76.95** | **76.54±0.318** |
    | PiSSA+AdaLoRA | **78.77** | **78.32** | **78.69** | **78.59±0.199** |

    **Table 2**: Fine-tuning DeBERTa-v3-base with PiSSA, LoRA (using Gaussian & Kaiming initialization), DoRA, and AdaLoRA on the GLUE benchmark, including 8 subtasks MNLI, SST-2, CoLA, QQP, QNLI, RTE, MRPC, and STS-B. The results for PiSSA, DoRA, and LoRA (with Kaiming initialization) are averaged over three runs. The other results, taken from the AdaLoRA paper, are averaged over five runs, with only the mean values reported.

    | Method | Run 1 | Run 2 | Run 3 | GLUE Average |
    | --- | --- | --- | --- | --- |
    | Full FT | — | — | — | 88.245 |
    | LoRA(gaussian-init) | — | — | — | 88.503 |
    | LoRA(kaiming init) | 88.795| 88.665 | 88.395 | 88.618±0.167 |
    | DoRA | 89.186 | 88.955 | 88.810 | 88.984±0.155 |
    | AdaLoRA | — | — | — | 89.464 |
    | PiSSA | **89.915** | **89.783** | **89.711** | **89.803±0.084** |

    **Table 3**. The gradient direction of the first step for PiSSA and LoRA initialized with five different random seeds, using the same batch of 128 data points, was reduced to two dimensions. LoRA's gradient direction is zero and random, while PiSSA's direction is consistent, related to the principal singular value of the model.

    |  | Method | Seed 0 | Seed 1 | Seed 2 | Seed 3 | Seed 4 |
    | --- | --- | --- | --- | --- | --- | --- |
    | grad_A | LoRA | [0,0] | [0,0] | [0,0] | [0,0] | [0,0] |
    |  | PiSSA | **[0,1]** | **[0,1]** | **[0,1]** | **[0,1]** | **[0,1]** |
    | grad_B | LoRA | [-0.992,  0.122] | [ 0.9525,  0.304] | [ 0.4587, -0.888] | [ 0.241,  0.970] | [ 0.036, -0.999] |
    |  | PiSSA | **[1,0]** | **[1,0]** | **[1,0]** | **[1,0]** | **[1,0]** |

   **Table 4**. The ratio of the distance moved towards the target point by PiSSA and LoRA after 5 update steps.

    | Metric | Method | Step 1 | Step 2 | Step 3 | Step 4 | Step 5 |
    | --- | --- | --- | --- | --- | --- | --- |
    | Loss | LoRA | **0.8881** | 0.7943 | 0.6598 | 0.6021 | 0.5538 |
    |  | PiSSA | 0.8884 | **0.6476** | **0.4568** | **0.3657** | **0.3346** |
    | ratio_to_target_A | LoRA | 0.00% | 2.57% | 6.28% | 10.84% | 15.94% |
    |  | PiSSA | **4.31%** | **11.29%** | **17.30%** | **22.77%** | **27.71%** |
    | ratio_to_target_B | LoRA | 3.31% | 9.66% | 15.75% | 21.51% | 27.00% |
    |  | PiSSA | **4.29%** | **11.27%** | **17.29%** | **22.77%** | **27.72%** |

    **Table 5**. Using a normal distribution to fit the original model and the residuals model in PiSSA, then recoding the mean and standard deviation of the distribution.

    |  | mu | sigma |
    | --- | --- | --- |
    | LLaMA-2-7B | 5.4679e-06 | 0.0193 |
    | PiSSA-LLaMA-2-7B-r128 | **2.6775e-06** | **0.0172** |
    | Mistral-7B | 9.8105e-07 | 0.0033 |
    | PiSSA-Mistral-7B-r128 | **5.3738e-07** | **0.0029** |
    | gemma-7b | 1.3422e-06 | 0.0045 |
    | PiSSA-gemma-7b-r128 | **6.9983e-07** | **0.0040** |
    | LLaMA-3-8B | 5.6969e-06 | 0.0139 |
    | PiSSA-LLaMA-3-8B-r128 | **2.3858e-06** | **0.0118** |

---

### Decision · Program_Chairs · 2024-09-25

**Decision:**

Accept (spotlight)

**Comment:**

The paper  presents a novel approach to parameter-efficient fine-tuning (PEFT) of large language models (LLMs). The proposed method, PiSSA, is an enhancement of the existing LoRA (Low-Rank Adaptation) method. PiSSA differentiates itself by initializing the adaptation matrices with the principal components obtained through SVD of the original model weights, as opposed to LoRA’s random initialization. This modification allows for more efficient fine-tuning. Additionally, PiSSA shows reduced quantization error when combined with 4-bit quantization (QPiSSA), making it a superior choice for resource-constrained environments.
There were concerns related to the clarity of the presentation and the  statistical significance of the results, both of which the authors have addressed in their rebuttal. Given the novelty of the approach and its strong empirical performance, I recommend the paper for acceptance with a revision to address concerns raised through the reviewer discussions.